



# Nitrogen cycling in the Elbe estuary from a joint 3D-modelling and observational perspective

Johannes Pein[1], Annika Eisele[1], Richard Hofmeister[1], Tina Sanders[1], Ute Daewel[1], Emil V. Stanev[1], Justus van Beusekom[1], Joanna Staneva[1], Corinna Schrum[1]

[1]Institute of Coastal Research, Helmholtz-Zentrum Geesthacht, Geesthacht, 21502, Germany

*Correspondence to*: Johannes Pein (johannes.pein@hzg.de)

**Abstract.** The study addresses the nitrogen cycling in Elbe estuary. Observations of salinity, nutrients and oxygen from moored stations, ship casts and helicopter surveys are presented. Observations are complemented by simulations obtained from a coupled physical-biogeochemical 3D unstructured model, applied for the first time to the estuarine environment. Model

simulations reproduce the temporal variability of nutrients and oxygen along the estuarine salinity gradient. Both, observations and model results, demonstrate mostly conservative mixing of nitrate and non-conservative behavior of ammonium. Model hind-casts of the years 2012 and 2013 provide a detailed reconstruction of nitrogen recycling with ammonium appearing as the key species of the remineralisation process. Estuarine turnover processes are fueled by inputs of diatoms and organic nitrogen at the tidal weir with intense primary production manifest in the shallow river section downstream of the weir. The

harbor area is the hot spot of heterotrophic decay associated with growth of meso-zooplankton, sedimentation of degradable material, remineralisation, oxygen depletion, denitrification and ammonium production. In the harbor, biochemistry shows strong vertical gradients while hydrodynamics demonstrate connectivity between the main channel and the harbor. At the estuary bed nitrogen is deposited during spring and early summer. Resuspension leads to nearly closed budget by the end of the year. During the Elbe flood in June 2013, estuarine biogeochemistry is significantly disturbed with the harbor being

deactivated as hot spot of heterotrophic decay. Plankton and organic matter are flushed towards the outer estuary which in consequence sees high abundance of grazers, oxygen depletion and elevated release of ammonium.

## 1 Introduction

The Elbe estuary is the largest estuary of the German Bight and has been subject to numerous observational and modelling studies during the last decades (Schuchardt and Schirmer, 1991; Kappenberg et al., 1995; Stehr et al., 1995; Schroeder, 1997;

Van Beusekom et al., 1998; Goosen et al., 1999; Rolinski, 1999; Kerner and Spitzy, 2001; Kerner, 2007; Dähnke et al., 2008; Schlarbaum et al., 2010; Amann et al., 2012; Sanders et al., 2018). It is a major driver of ecological change in the German Bight and adjacent Wadden Sea (e.g. Niermann et al., 1990; van Beusekom et al., 2009; Emeis et al., 2015). The continuous historical anthropogenic changes leading to regime shifts in the estuarine ecosystem have additionally fueled the scientific interest into the Elbe estuary (see overview in Pusch and Fischer, 2006; Kerner, 2007; Holzwarth and Wirtz, 2018). The



estuarine physics have been shaped by coastal protection measures and diking (Mikhailova, 2011) and dredging (Rolinski and Eichweber, 2000). They will continue to change given further plans to deepen the navigational channel (Jickells et al., 2014). The estuarine biogeochemistry couples strongly to processes taking place upstream of the tidal weir and comprise large scale interferences such as leaking of nutrients due to agricultural activities in the catchment area (Dähnke et al., 2008). Hence the Elbe ecosystem depends on political processes and resulting economic and environmental policies to a yet unknown degree.


The Elbe has developed from a meso-trophic environment into a highly eutrophic environment with the onset of industrial agriculture (Pusch and Fischer, 2006). Here we want to recall shortly the general development preceding the current environmental state of the estuary: 1) 1970s-1980s: High pollution limited phytoplankton growth. At the landward boundary high ammonium and phosphate loads entered the estuary while denitrification of nitrate happened in estuarine sediments; 2)

1990s: The pollution decreased due to demolition of Eastern German industries and progress in wastewater treatment (Sanders et al., 2018). Ammonium concentrations decreased. Nitrate manifested pseudo-conservative mixing. The problem of summer oxygen depletion appeared; 3) since 2000: Nitrate dominates the estuarine nitrogen species and observations indicate its conservative mixing (Dähnke et al., 2008). The river part sees most intense phytoplankton growth with biomass reaching a maximum of around 150 µg Chlorophyll at the tidal weir (Scharfe at el. 2009; Schöl et al., 2014).


The oxygen and nutrient dynamics in the tidal river have recently been successfully addressed by several models. Schöl et al (2014) have presented an offline-coupling of a water quality model to a depth-averaged hydrodynamical model. These authors were able reproduce multi-year dynamics of plankton, oxygen and nutrients along the freshwater part of Elbe river and Elbe estuary until Cuxhaven. Geerts et al. (2017) introduced a joint study of the Schelde and the Elbe estuaries using 1D

representations of the respective areas. Focusing on the estuarine oxygen budget, they successfully predicted the overall dissolved oxygen dynamics relying on a simple balance between biological oxygen demand (caused by decaying organic matter) and oxygen production (through re-aeration). Without resolving primary production or specific nutrient pools, they were able to show that the oxygen budget depends critically on the estuarine geometry. The model of Holzwarth and Wirtz (2018) was used to forward similar conclusions increasing however the model complexity. Their prediction tool was a NPZD

model truncated off the secondary production (the "Z") but with a more detailed resolution of the organic matter decay. Coupling with hydrodynamics was off-line, too. The model successfully resolved the biogeochemical gradients between the limnic zone and the salinity front reproducing in particular the Elbe summer oxygen minimum.

The above studies showed that even with spatially and biogeochemically simplified models it is possible to coherently simulate

major estuarine processes and answer question regarding climate impacts on estuaries or estuarine management. These studies however did not resolve the entire transition between the tidal weir and the oceanic background while assuming vertical (and some even lateral) homogeneity of the gradient-rich estuarine environment. We hypothesize that such simplification is not justified. Here we show that a highly-resolved 3D hydrodynamical model of the Elbe estuary coupled on-line with a

biogeochemical model of decent complexity 1) reproduces the seasonal and along-channel variability of major biogeochemical

state variables and 2) reveals lateral and vertical gradients largely unknown to the scientific community so far which 3) lead

to a refined understanding of the Elbe estuary ecosystem that diverges from the traditional 1D description. The study focusing

on nitrogen cycling, presentation of observations and model results comprises the state variables controlling nitrogen

transformation, i.e. the inorganic and organic nitrogen species, silicate, oxygen and plankton. The overall aim is to describe

the pathways of nitrogen inputs at weir towards they ocean and possibly vice versa. In detail we will tackle the following

research questions: 1) Are riverine nitrogen inputs completely transformed to nitrate; 2) Does the finding of conservative

mixing of nitrate hold on seasonal and interannual scales, 3) What is the role of organic nitrogen; 4) how stationary are process

of the nitrogen cycling; 5) how is the timing of their spatial variability. Special focus is dedicated to the Elbe flood event in

2013 which represents a naturally driven perturbation experiment, challenging the resilience of the estuarine ecosystem.

## 2 Methods

Long-term data of nutrient concentrations and chlorophyll-a were obtained at different stations along the Elbe estuary, namely

weir Geesthacht (km 585.9), Zollenspieker (km 598.7), Hamburg (km 630.1; since 1982: km 628.8), Grauerort (km 666.5;

since 1986: 662.7; since 1988: 660.5), Brunsbüttel (km 693.0) and Cuxhaven (km 725.2). Measurements were taken by FGG

Elbe as individual samples or horizontal profile mixing probes one hour before low tide. Station data are available from 1979

– 2014 on a daily basis measured once per week to once per month, but data gaps exist to different extents depending on

parameter, measurement station, season and year. These observations served as boundary conditions to force the model

simulations presented herein.

Transect data of nutrient concentrations and salinity for the lower part of the Elbe river were sampled during Helicopter flights

(brown circles in Figs. 8, 9) and ship surveys (blue circles, Figs. 8,9; see Sanders et al., 2018 for details). Sampling by helicopter

allowed to complete a single survey of the estuary during the same tidal phase. Samples were taken on average every ~3.7 km

along the river from 1980 – 2014 four to twelve times a year.

All nutrient measurements were analysed by the same laboratory using a limnic method for all stations, except Cuxhaven for

which a marine method was applied. Therefore nutrient values at the lower (seaward) end of the estuary could be slightly too

high.


The estuarine nutrient budget critically depends on discharge. River runoff was measured daily at the last tide-free gauge

located at Neu Darchau (km 536.4). At this station, the annually averaged freshwater runoff is 717 $m^3$ $s^{-1}$ (SD: 459 $m^3$ $s^{-1}$,

years: 1979 – 2014). Precipitation and melting water lead to highest discharges in April (climatological mean: 1112 $m^3$ $s^{-1}$;

SD: 508 $m^3$ $s^{-1}$) and lowest in September (climatological mean: 439 $m^3$ $s^{-1}$; SD: 187 $m^3$ $s^{-1}$). The discharge climatology shows

highest variability from January until April with 1990s discharge values being extremely low in these months. To investigate



estuarine nitrogen transformation processes river discharges were multiplied with nutrient concentration observations to derive loads of nitrate ($NO_3^-$), ammonium ($NH_4^+$), phosphate ($PO_4^-$), silicate ($SiO_4^-$) and organic nitrogen.

## 3 Model

The hydrodynamical core of our Elbe-model consists of the SCHISM (Zhang and Baptista, 2008; Zhang et al., 2016) applied
to the year 2010 topography (downloaded from www.portal-tideelbe.de). In the recent years, SCHISM has been successfully applied for the estuarine modelling (Pein et al., 2014; Ye et al., 2018; Liu et al., 2018; Stanev et al 2019). For simulation of the Elbe estuary we adopt the necessary techniques, such as vertical and horizontal scale resolving coordinates on an unstructured mesh and sigma coordinates, the drying and wetting algorithm, TVD advection scheme, 2 equation turbulence model, parallel implementation to run on supercomputers. Central to the focus of this study, which is nitrogen cycling, is the
coupling between the hydrodynamical core and a biogechemical model. As interface between the two models we use the FABM framework (Bruggemann and Bolding, 2014), which couples ecosystem models to integrate at the same timestep and using the same advection scheme as the physical tracers. The technical configuration of the estuarine biogeochemical simulator is a subset of the GCOAST model tools at the Helmholtz-Zentrum Geesthacht.

**3.1 Physical set-up**

The computational triangular mesh resolves the model area (Fig. 1) with approximately 33k mesh nodes. Mesh resolution near the open boundaries equals 500 m. Between Cuxhaven and Geesthacht the channel resolution varies with the channel width. A minimum of eight cells covers the estuarine channel in the lateral direction while the length-to-width ratio of the mesh cells equals roughly 3:1. In the vertical the model mesh has 21 s-coordinates. At the open ocean boundary sea level, horizontal
currents, salinity and temperature are constrained by hourly outputs from an unstructured model of the German Bight (Stanev et al., 2019) that has been integrated for the period of 1 January 2012 until 31 December 2013. Daily discharge measurements by the German waterways agency (WSV, www.portal-tideelbe.de) stipulate the Elbe river freshwater contribution to the Elbe estuary. Atmospheric variability is represented by precipitation, wind, atmospheric pressure and solar insulation coupling the Elbe model to hourly reanalysis data provided by the German Weather Service (DWD COSMO-EU model with 7 km spatial
resolution).

**3.2 Biogeochemical model and forcing**

ECOSMO2 (Daewel and Schrum, 2013; Schrum et al., 2006) was originally developed as a generic robust lower trophic-biogeochemical model for different shelf sea systems, with the focus to North Sea and Baltic Sea. The model accounts for the primary production of three functional phytoplankton groups, predation by two functional zooplankton groups and nutrient
cycling in the water column as well as exchange processes between water and sediment. In the deeper part of Elbe estuary





heterotrophic processes prevail over primary production (Schöl et al., 2014; Holzwarth and Wirtz, 2018). Therefore model formulation for organic matter decomposition is of utmost importance.

The model assumes the Redfield ratio for C, N and P. Although model equations are written in units of mg C m$^{-3}$, we will

focus in the following on nitrogen keeping in mind that in the model 12 mol N correspond to 106 mol C. Nitrogen enters the model either through plankton, as inorganic nutrient (NO3 or NH4) or dissolved or particulate matter (detritus). It is removed from the water column either by denitrification of NO3 or by descending of detritus into the sediment pool. The latter process is reversible by resuspension. The organic particulate matter does not feedback onto the physical system. The organic sediment represents a mere storage of nutrients to the biological system. Depending on oxygen conditions release of nitrogen from the

sediment surface results either in formation of nitrate by nitrification (oxic) or release of ammonium (hypoxia with oxygen concentration below 120 mmol m$^{-3}$ and anoxic). Mineralisation of organic matter in the sediment pool leads to increasing ammonium and phosphate concentrations in the lowermost pelagic cell. Mineralisation under denitrification conditions uses oxygen and nitrate from the lowermost pelagic cell generating near-bottom oxygen and nitrate depletion. The attenuation of light within the water column is calculated based on a background attenuation and a linearly increasing coefficient depending

on plankton concentration. In effect light attenuation is controlled predominantly by water depth, which has been shown a valid approach for the area of Elbe estuary in the study of Holzwarth and Wirtz (2018).

A sponge layer is applied to the open ocean boundary where model concentrations of nitrate, ammonium, silicate and particulate organic matter are nudged to oceanic time series derived from observations provided by ICES (www.ices.dk). The

oceanic forcing integrates data from the southern North Sea (ICES area "IVb") which is used to derive signals with sufficient resolution of the seasonal variability. The sponge layer extends approximately 5 km from the open ocean boundary into the model area interior. The nudging drags respective model variables towards observations depending on a relaxation constant specified within the sponge layer. If the relaxation constant equals unity, which is the case at the outer edge of the sponge layer (identical with the open boundary), the model is fully relaxed towards observations in this region. A relaxation constant of 0.5

means that the local model output equals the average between the value predicted within the model and the observed value. From the outer edge of the sponge layer relaxation constants decrease constantly reaching zero at its inner edge. Here the model works freely. The particulate organic matter was estimated as the difference between observed total nitrogen minus observed inorganic nitrogen. Primary and secondary production, oxygen and opal are not nudged to observations however they respond to the nudged within the sponge layer variables. Very relevant for the forcing of the estuarine ecosystem from the land

side are discharges of water and nutrients at the tidal weir:  At Geesthacht we apply observed by WSV water discharge (m$^3$ s$^{-1}$). Freshwater inflow is assigned observed concentrations (see above Sec. 2) of nutrients and oxygen (mmol m$^{-3}$). The nutrient load entering the estuary from the landside thus equal the product of discharge and nutrient concentration (mmol s$^{-1}$). Main contributor to organic nitrogen at the tidal weir are diatoms (e.g. Schöl et al., 2014). We derived diatom concentrations (mmol N m$^{-3}$) from measurements of chlorophyll-a using the ratio of 1.58 g chl-a per mole nitrogen (Cloern et al., 1995; Zhou et al.,



2017). We assume that small portions of meso-zooplankton and small phyto-plankton (both with ratio of 1:10 to diatoms) also enter the estuary at Geesthacht. Non-living labile organic nitrogen was estimated from measurements of total nitrogen with inorganic nitrogen and an assumed 45 mmol m$^{-3}$ of terrestrial refractory organic nitrogen substracted. The estimated organic nitrogen enters the estuary at the tidal weir as 'dissolved organic matter' in the definition of ECOSMO2 (Daewel and Schrum, 2013). Spin-up time of the model was one year.

## 4 Validation and physical background

### 4.1 Tides and salinity front

The physical setting of the estuarine eco-system is dominated by tidal motion and the estuarine salinity gradient. Their adequate resolution is crucial for reproducing the interaction of physical and biogeochemical systems. In order to assess the realism of the tidal simulation, we performed a tidal analysis of both observed and simulated sea levels along the Elbe estuary using the

UTide software (Codiga, 2011). The comparison (Fig. 2a) of observed and simulated M-2 tide demonstrates good performance of the model with deviation of water levels smaller than 10% between Hamburg and Cuxhaven (for locations see map Fig. 1). This is an important result confirming realistic ventilation of the estuarine main channel by tides.

For validation of simulated estuarine salinity gradient, we used salinity measurements obtained during ship-borne CTD casts

described in Sec. 2. Model salinity plotted against observed salinity displays a  scatter of points along the diagonal during April, July and August cruises in 2013 (Fig. 2 c-e) with model errors no greater than 5 psu. Model salinity grossly overestimated the observed during October 2012 salinity. Overall, the model revealed a tendency to exaggerate salinity intrusion, especially at low river discharge (Fig. 2b, c vs. black dashed line in Fig. 3c). Model temperature was smaller than observed temperature, revealing a bias of approximately 2 °C (Fig. 2 b-e). With the exception of October 2012 (Fig. 2b), sea surface temperatures

demonstrated an estuarine gradient (Fig. 2c, d, e). Model performance was slightly better in the landward direction (Fig. 2c). The time-versus-distance diagram of sea surface salinity (Fig. 3a) gives a more complete picture of the estuarine salinity variability. In 2012, estuarine salinity followed a typical cycle of retreated towards the sea steep salinity front in winter and advancing into the estuarine shallower salinity front during summer. In the year of 2013 this trend was perturbed by a major flood event in June (see Voynova et al., 2017). In this month of extreme river discharge, the salinity front retreated towards

the ocean as far as during winter periods in both 2012 and 2013. This flood event constitutes thus a natural 'experiment' of its own demonstrating the functioning of the estuarine ecosystem during the biologically active period of the year under weakened influence from the ocean.

While salinity traces the advection and distribution of the biogeochemical constituents, certain biological processes such as

remineralisation depend on the water temperature. A time-versus-distance diagram of the sea-surface temperature illustrates the annual cycle during 2012 and 2013 (Fig. 3b). Other than salinity, temperature features small along-channel gradients in



comparison with the temporal variability. It can be seen however that during spring and summer the estuarine temperature first rise near the head of the estuary, the heating subsequently propagates towards the lower reaches. Overall the first year of integration, 2012, appeared to be colder than the second year, 2013, with colder winter minimum and colder summer maximum
(Fig. 3b).

The combined effect of tidal and salinity forcing interaction with the channel topography accounting for along-channel dispersion of tracers (Dyer, 1973; Helder and Ruardij, 1982). A simple estimate of the along-channel dispersion coefficient (Steen at al., 2002) reads

$$K(x) = \frac{\overline{U*S}}{dS/dx},$$ (1)

where $S$ is salinity, $x$ is the along-channel coordinate and $U$ is the along-channel velocity component. Steen et al., 2002 derived $U$ from the ratio of river discharge $Q$ and the cross-sectional area of the channel $A$. We assumed $U$ to be the horizontal surface velocity magnitude, taking into account that the interaction of both vectors controls along-channel dispersion (Fischer et al., 1979).


The estimated along-channel dispersion revealed an intrigue relationship with the river discharge (see yellow solid line compared against black dashed line in Fig. 3c): At low river discharge, as during June and September 2012, the estimated along-channel dispersion at Cuxhaven (Elbe-km 725) reached 70 $m^2$ $s^{-1}$ and 90 $m^2$ $s^{-1}$, respectively. At high river discharge such as in February and March 2012 the estimated along-channel dispersion in the same location reduced to less than 5 $m^2$ $s^{-1}$.
This makes an important difference because the along-channel dispersion is a bi-directional process enabling upstream transports against the direction of mean flow. We illustrate its possible relevance for the estuarine biogeochemistry by plotting time series of mean tracer age concentration at Cuxhaven of two age tracers (Fig. 3c), where the blue solid line represents an age tracer entering the model area at Geesthacht (for the location see Fig. 1) and the red solid line accounting for an age tracer entering the model area at the seaward open boundary. The river-borne age tracer correlated with the along-channel dispersion
while anti-correlating with the river discharge (Fig. 3c). In March 2012 the average tracer age near Cuxhaven was ~ 30 days while during July and October 2012 it reached 80 days. The second age tracer showed less obvious relation to river discharge. Correlation of the tracer concentrations with estimated dispersion coefficient was R=0.91 and R=0.31 for the river-borne and the sea-borne age tracers, respectively. In June 2013 river discharge exceeded 4000 $m^3$ $s^{-1}$ ( Fig. 3c). This immediately reduced local estimated dispersion and residence time, which was then only ten days for river-borne water. Increased residence times
would occur for both river-borne and oceanic water parcels at low discharge and increased dispersion. On the other hand topographic features would affect horizontal velocities and horizontal salinity gradient locally, controlling thus horizontal dispersion. Average along-channel tracer age concentration interacted locally with the average along-channel dispersion (red solid line and yellow solid line around Elbe-km 732 in Fig. 3d), revealing elevated upstream mixing of 'younger' oceanic water due to locally increased dispersion. The river-borne water showed slight steepening of the along-channel gradient in the

same location and thus opposite response (blue solid line in Fig. 3d at Elbe-km 732). The complex nature of the underlying processes controlling residence times and tracer dispersion (with biogeochemical turnover happening on top) justifies the use of a complex 3D model to simulate estuarine tracer transports because such a model accounts for interaction of tides and freshwater discharge, salinity and topography that cannot be considered in an isolation.

## 4.2 Nutrients

Our validation of the Elbe estuary ecosystem simulation focuses on nutrient and oxygen concentrations. Dissolved nutrients provide observable conditions of the ecosystem state. Oxygen concentrations depend on the local balance between primary production, respiration and reaeration. In general, we find that the model adequately represents the along-channel differences of the inorganic nitrogen species, both important nutrient and product of the nutrient recycling. Model results show particularly small deviations from the seasonal dynamics of silicate and oxygen, whereas the former is a basic nutrient to diatoms and the

latter displays the balance between production and respiration. The models appears thus as an appropriate tool to address the specificities of nitrogen cycling in the Elbe estuary. The validation is organized in two parts: 1) Validation of inorganic nitrogen species, oxygen and silicate against stationary measurements from the same provider as the data used for forcing at the tidal weir (see a description of long-term stationary data in Sec. 2); 2) validation of the two inorganic nitrogen species against the salinity gradient against data from both ship surveys and helicopter casts (see Sec. 2).


Figs. 4, 5, 6 and 7 illustrate the temporal variability of nitrate, ammonium, oxygen and silicate, respectively, at six stations from the tidal weir to the outer estuary (see. Fig. 1 for names and positions) during the two years of integration as seen by observations and model. Figs. 4a, 5a, 6a and 7a reflect the prescribed by the observations model forcing at the tidal weir. Naturally away from the weir signals experienced modification seeking explanation by either local processes or mixing with

downstream waters.

Nitrate constitutes the major nitrogen species entering the estuary. Concentrations at the weir may be modified downstream by nitrification of ammonium, consumption by primary and secondary producers and denitrification in the water column or in the sediment. Nitrate demonstrates a clear annual cycle with high concentrations in winter and low concentrations in summer

(Fig. 4a). Figs. 4a-e reveal that winter concentrations remain constant between the tidal weir and station Brunsbüttel, before relaxing towards oceanic concentrations at station Cuxhaven (Fig. 4f). In summer, concentrations at the weir fell below 70 mmol m$^{-3}$ which roughly equaled concentrations in the lower reaches (Figs. 4a, f). Between stations Grauerort (Elbe-km 663) and Brunsbüttel (Elbe-km 693), nitrate concentrations rose to ~110 mmol m$^{-3}$ indicating a nitrate source within the estuary. The model reproduced these general temporal and spatial trends with two notable modifications: 1) At stations Cuxhaven and

Brunsbüttel model nitrate revealed a strong spring-neap variability that appears upstream at Hamburg during the summer months. 2) The model overestimates summer nitrate concentrations at Hamburg and even more so at Grauerort (Figs. 4c, d).



Since simulation performed very well outside the section Hamburg-Grauerort the explanation to this deviation must be local processes such as denitrification.

Ammonium has only little annual variability at the tidal weir (Fig. 5a). Observations indicate winter low and summer high concentrations at station Hamburg (Fig. 5c) whereas at Grauerort the annual maximum is attained in early spring (Fig. 5d). While reproducing 60-70 % of the magnitude of the ammonium concentration peaks, the model suggests a stronger seasonal fluctuation at the three stations between the harbor and the position of the estuarine turbidity maximum (ETM) than the observations (Fig. 5c-e, Fig. 1; for position of ETM see Stanev et al., 2019; Kappenberg and Fanger, 2007). At Cuxhaven

station observations indicate a bi-modal annual signal with peak concentrations of around 10 mmol m$^{-3}$ (Fig. 5e). The model predicted several peaks in both years of integration. These peaks attained magnitudes of 7-8 mmol m$^{-3}$. Overall, the model thus reproduced the seasonal amplitude of ammonium concentrations and the spatial differences between stations as seen by the observations. However, a point worth discussion seemed the underestimation of winter concentrations by the model at and downstream of Hamburg (Figs. 5c, d).

Oxygen levels represent an important benchmark for biogeochemical modelling in estuaries because oxygen is central to the transfer of matter from organic pool to inorganic pool, and vice versa. For comparison, both observations of oxygen concentration and modelled oxygen have been converted to oxygen saturation, using observed and simulated thermohaline data, respectively. At the two most upstream stations oxygen showed a clear annual cycle with summer highs und winter lows

in 2012 (Figs. 6a, b). The same pattern largely repeated in 2013 with the exception of a prominent negative peak during the aforementioned Elbe flood event in June 2013. At Hamburg and Grauerort stations, the annual signal seen at the upstream stations basically changed sign revealing oxygen depletion during summer and roughly one hundred percent saturation during winter (Figs. 6c, d). Towards Brunsbüttel station summer depletion weakened and Cuxhaven station saw mostly oversaturated oxygen conditions (Fig. 6e, f). The model reproduced observed annual oxygen dynamics with high accuracy (Fig. 6). In

particular, it tackled well the transition from autotrophic to heterotrophic regime between Grauerort and Hamburg (Fig. 6b, c). Moreover, the model predicted fortnightly fluctuations during summer, which was also manifested by the ammonium, and partially nitrate concentrations (Figs. 4, 5). At the most seaward station, the model systematically under-estimated oxygen saturation.

The silicate loads entering the Elbe estuary at the tidal weir exert a strong seasonal forcing on the downstream biogeochemical processes. In the shallow part of the estuary, the stations reveal an almost rectangular signal with winter high of 115-130 mmol m$^{-3}$ and summer low between zero and several mmol m$^{-3}$ (Fig. 7a-c). In the deep part of the estuary, winter silicate levels decrease towards the German Bight (Fig. 7e, f). Summer concentrations increase at the same stations, indicating relevance of mixing between ocean and riverine water in the respective estuary section in both seasons. Simulated silicate concentrations

follow closely the observed levels in the shallow part of the estuary (Fig. 7a, b, c) with some underestimation in the beginning





of the year 2012 at stations Zollenspieker and Hamburg (Fig. 7b, c). The model accurately reproduces the timing and the amplitude of the observed seasonal variability of silicate. At stations Hamburg, Brunsbüttel and Cuxhaven the model tends to underestimate the summer silicate levels (Fig. 7 c, e, f), however the mismatch appears small (< 15 %) compared to the seasonal signal amplitude.

### 4.3 Nitrogen-salinity relationships

To complete the validation section we present mixing plots of inorganic nitrogen species generated from model and – in case of availability – observations. The idea of plotting a biogeochemical property against salinity is to identify conservative or non-conservative behavior of the property along the estuarine density gradient (see for example Sharp et al., 1986).

Both model results and observations – the latter from ship and helicopter surveys - revealed mostly conservative mixing of nitrate at salinities greater than 10 psu during spring, summer and autumn (Figs. 8a-c). The regression lines of model nitrate and observed nitrated showed the same trend, indicating very similar mixing rates across the estuarine density gradient. At lower salinities, convex mixing curves implied a source of nitrate in the upper reaches in spring (Fig. 8a). At this time of the year, nitrate levels increased from ~50 mmol m$^{-3}$ at the weir to more than 200 mmol m$^{-3}$ at salinities between 0 and 1 psu (see the same increase of nitrate concentrations from Geesthacht to Grauerort around 1 May 2012 in Fig. 4a, c). In summer, the mixing curves conserved their convex shape while clearly extending the conservative regime towards the limnic zone (Fig. 8b). At this period of the year, the model overestimated limnic nitrate generation compared to the majority of measurements or loss by denitrifications or sedimentation of particulate matter bound N was underestimated. In autumn, observations revealed concave bending at low salinities, indicating a nitrate sink in this area (Fig. 8c) or gradually increasing nitrate concentrations in the river Elbe (Fig. 4a). Model results scattered considerably at the low salinities implying a transition between the two regimes of nitrate production and consumption. Regression analysis of model results and observations showed close agreement (give statistics) and thus good realism of the model with respect to the nitrate-salinity relation. During the winter period, no ship- or helicopter-borne observations were available which is why data from observation Brunsbüttel und Cuxhaven stations are shown in Fig. 8d. Model mixing plots showed basically two regimes: 1) strictly conservative mixing at the time of the highest nitrate inputs in February-March (Figs. 8d, 4a), 2) slight consumption of nitrate at low salinities similar to the autumn situation (Fig. 8d compared to Fig. 8c) at nitrate concentrations near the weir of ~ 250 mmol m$^{-3}$ (see 1 January in Fig. 4a).

Mixing curves of ammonium demonstrated non-conservative mixing with concave distribution over salinities below 15 psu of the scatter points from both observations and model during spring, summer and autumn (Fig. 9a-c). The mixing plots concealed the spatial variability of ammonium in the limnic zone that becomes obvious from Fig. 5. They emphasize however summer and autumn ammonium production at salinities between 15 and 30 psu. For the winter period no observations were available. Model results revealed almost complete inertia of the ammonium concentration against the estuarine density gradient with scatter arising from variability at the open boundaries (Figs. 9d, 5a).



## 5 Results and discussion

After assessing the validity of the coupled physical-biogeochemical simulations in the area of the Elbe estuary in the previous section, in this section we want to reveal the full spatio-temporal dynamics of the estuarine ecosystem based on the model simulations.

### 5.1 Seasonal and along-estuary ecosystem dynamics

Distance-over-time diagrams illustrate the biogeochemical dynamics along the Elbe estuary during the two years of model
integration (Fig. 10). The bottom line of the panels in Fig. 10 represents the forcing at the tidal weir. The seasonal forcing of inorganic nutrients has been described in the previous Section 4b (see also Figs. 4a, 5a, 6a, 7a). While inorganic nutrients dominated riverine forcing during winter, organic loads prevail during the productive time of the year. They are composed of a main living and a main non-living component – diatoms and dissolved organic matter (Fig. 10a, g). Diatoms revealing approximately 50 % higher maximum loads than dissolved organic matter (i.e. 120 mmol N m$^{-3}$ of diatoms compared to 60
mmol N m$^{-3}$ of dissolved organic matter), both inputs to the estuary must be considered crucial for downstream biogeochemical turnover. Interestingly the two signals are rather uncorrelated as manifested by the different timing of pulses of diatoms concentrations vs. pulses of dissolved organic nitrogen concentrations (3 vs. 1 in 2012, 2 vs. 3 in 2013).

Away from the weir, biogeochemical state variables either mix conservatively with the ocean water upon reaching the salinity
front or they experienced inner sources or sinks. The first case has been described in the previous section 4c for nitrate in particular during wintertime. With the spatial dimension fully resolved by the used unstructured-grid model, wintertime nitrate concentration appeared constant over the limnic part of the estuary and steeply decreased downstream from km 680 (Fig. 10d). Silicate behaved very similarly to nitrate reflecting low biogeochemical turnover at cold temperatures (Fig. 10f, compare Fig. 3b). As early as April first diatom blooms reached the estuary from upstream and the winter nutrient strongly decreased (Fig.
10a). The river blooms were associated with increased levels of oxygen (Fig. 10i) and dissolved organic nitrogen (Fig. 10g). In spring, concentrations of diatoms, oxygen and dissolved organic nitrogen stayed constant along the short stretch between the weir and the station Hamburg at Elbe-km 629, which is located just downstream of the Hamburg harbor area (see Fig. 1). Here, grazing occurred with meso-zooplankton levels showing a peak (Fig. 10c). At the same time ammonium and dissolved and particular organic nitrogen increased (Fig. 10e, g, h). In our model of the Elbe estuary grazing appeared as the main reason
for phytoplankton reduction, conforming the findings of Schöl et al., 2014 with a 1D model. Downstream from the area of intense grazing the concentrations of nitrate also increased (Fig. 10d), indicating nitrification of ammonium which is exactly the source of nitrate that was evident from the mixing plots (Fig. 8, see Sec. 4c). These spring dynamics were approximately the same in 2012 and 2013.



Summer dynamics revealed a couple of specific differences. The area densely populated by meso-zooplankton moved slightly downstream with its epicenter located in the harbor area (Fig. 10c). The secondary producers decimated the diatom concentrations, releasing ammonium, dissolved organic nitrogen and particulate organic nitrogen (fecal pellets) (Fig. 10e, g, h). The increased by the secondary producers' respiration reduced oxygen saturation to well below 80% (Fig. 10) and secondary producers at least partially account for the ammonium peak at and downstream of Hamburg harbor.


Holzwarth and Wirtz (2018) as well as Geerts et al. (2017) have emphasized the importance of the bathymetric step (between the shallow part of Elbe estuary and its deeper part, beginning in the Hamburg harbor) for the transition from the autotrophic regime to heterotrophic dominance. Our simulations reveal such a transition in the same area (as been from peak of meso-zooplankton concentrations, detritus and ammonium in Fig. 10 c, h, e) and seem to confirm the importance of the change in

channel geometry at Hamburg for the regime shift of the estuarine biogeochemical dynamics. We found however that the regime shift might not only me attributed to changed channel depth or vertical dimension only, and that a more complete understanding requires consideration of three-dimensional variability in time. At this point we like to remind we omitted re-parametrizing the equations in ECOSMOII model applied to North Sea and Baltic Sea by Daewel and Schrum (2013). We hypothesize that by coupling of a robust NPZD model including vertically integrated dynamically coupled sediment pools

(ECOSMOII), that was successfully applied to various dynamically different shelf and ocean regions (Daewel and Schrum, 2013; Holt et al., 2016; Samuelsen et al., 2019) with an unstructured 3D hydrodynamical model (SCHISM) specific calibration becomes unnecessary. The coupled system generically predicts the main spatio-temporal dynamics required to reproduce the specific estuarine biogeochemical dynamics. The successful validation of modelled dynamics in the harbor area (Sec. 4b) support this hypothesis.


An across-basin profile illustrates the average summer conditions in a major harbor basin, *Vorhafen* (see inset with harbor topography in Fig. 1), and adjacent river cross-section (Fig. 11, see for position of transect Fig. 1). Hydrodynamically, the harbor basin and the river section were clearly separated by the former revealing background levels of turbulent kinetic energy while the river experienced strong vertical mixing (Fig. 11b). Despite this sharp physical segmentation northern velocities

showed vertical shear prompting average transport from the river into the basin at mid-depth (upper black arrow in Fig. 11a) and vice-versa near the bottom (lower black arrow in Fig. 11a). Residence time in the basin showed tracer age concentration at river levels in the northern part of the basin and towards the surface (km 2, depth = 5 m in Fig. 11c). Towards the bottom and even more so inside the basin tracer age rose form ~ 7days to more than 10 days, with the oldest water aged ~ 12 days occurring at the bottom of the shallower southern part of the harbor basin (km 0 to km 1 in Fig. 11c). The lateral structure and

vertical stratification manifested by the age tracer was reflected by the biogeochemical variables associated with the heterotrophic chain: Grazer concentration increased from the river into the basin, with maximum concentration at the southern end of the section (Fig. 11d). Diatoms, prey for meso-zooplankton, showed an opposite gradient (contour lines in Fig. 11d), suggesting that grazers depleted diatoms inside the basin. Inside the basin, grazers' concentrations decreased near the basin





bed. Here, accumulation of particulate organic material (pellets and dead plankton) fueled the remineralisation of ammonium

and oxygen depletion (Fig. 11f, g, h). The low oxygen levels prompted denitrification such that nitrate concentration were lowest inside the harbor basin near the bed (Fig. 11e).

The specific interactions of these state variables became clearer from time series of their concentrations at a location inside the *Vorhafen* harbor basin (Fig. 12a, b; position identical with km 0 in Fig. 11; see also Fig. 1 with inset showing harbor structures):

The high winter concentrations imposed onto the estuarine dynamics at the tidal weir (Fig. 4a) showed opposite trends to ammonium and dissolved organic matter in 2012 (Fig. 12a). These were associated with strong zoo-plankton growth in the water column (Fig 12b) which was dynamically kick-started by the spring bloom of diatoms (Fig. 12b blue solid line shows however the sum of primary producers and micro-zooplankton). During the early spring bloom 2012 autotrophy dominated in the inner parts of the harbor but consumptive processes dominated from May on. The accumulation of zooplankton in the

harbor enhanced excretion of fecal pellets and subsequent sedimentation (Fig. 12b, c). The average sedimentation rates in the harbor were approximately twice as high as in the total area comprised by the section Zollenspieker-Hamburg (including the harbor) and the section Hamburg-Grauerort (see dashed red line vs. solid red and yellow lines in Fig. 12c). This emphasizes the importance of reduced hydrodynamics in harbor basins for local sedimentation and remineralisation. Sedimentation in the harbor has its seasonal maximum in end of May 2012. During summer, zoo-plankton concentrations, remineralisation of

ammonium and oxygen depletion increased reaching saturation levels as low as 10 % averaged over the water column (Fig. 12b, in the bottom cell anoxic conditions were reached (not shown)). The zoo-plankton and associated detritus appeared as the main agents controlling respiration. The zoo-plankton exceeded twice the maximum mass concentration of its prey and organic matter and is clearly anti-correlated with the oxygen saturation. The ventilation by tides brought diatom-rich waters from the Elbe main channel into the harbor basins such that zoo-plankton growth was sustained throughout the year (Fig. 12b). The

hydro-biogeochemical-dynamics controlled oxygen depletion and release of ammonium with its subsequent nitrification to nitrate reached from the harbor area down to the tip of the salinity intrusion. This is in line with in situ studies reaching the same conclusion (Sanders et al., 2018) and with distribution patterns of ammonium concentration support this finding (see Fig. 9b, c and supplementary figure S-2 b, c).

The time-versus-distance diagrams of meso-zooplankton and ammonium reflected the transition from spring bloom dynamics to the summer regime in 2012 (Fig. 10c, e): The harbor area appeared as an attractor of the respiration chain. The oxygen levels downstream from the ammonium maximum recovered, enabling nitrification and leading to the summer estuarine nitrate maximum stretching between Elbe-km 640 and 690 (Fig. 10 I, d) that marked the beginning of the conservative mixing regime during this period of the year (Fig. 8b). The nitrate maximum was thus located around the 10 psu isohaline (Figs. 10d, 2a). The

fact that the nitrate maximum is not at the start of the salinity gradient but downstream suggests that not all riverine organic matter was degraded when reaching the salinity gradient. Alternatively, the nitrate maximum at 10 psu may be partially due to degradation of marine organic matter.





Extrapolating the nitrate-salinity relation from salinities higher 10 psu to salinities lower than 10 psu would approximately
predict the total concentration of nitrogen at 0 psu (e.g. Officer, 1979). In Fig. 8 the dashed lines represent the regression lines
of the model nitrate-salinity relation at salinities greater than 10 psu. The blue crosses mark the average total nitrogen inputs
at the weir during the seasons represented by the four panels in Fig. 8 (spring: April to June, summer: July to September,
autumn: October to December, winter: January to March). During spring, autumn and winter model results supported the
hypothesis of complete nitrogen conversion into nitrate in the limnic and low saline reaches (Fig. 8a, c, d). In summer on the
contrary, the riverine inputs of nitrogen were well above the nitrate concentrations at 0 psu. In fact at this period of the year,
the estuarine ecosystem developed a more stretched cascade of the nitrogen turnover due to 1) sedimentation in the section
Zollenspieker-Hamburg which was positive in August and September 2012 (Fig. 12c) and 2) because of zoo-plankton growth
(Fig. 10c). Both processes made nitrogen unavailable for nitrification, slowing down the remineralisation chain with nitrate as
the end-product for export into the ocean. Although buffering of nitrogen in terms of rates was dominated by the harbor area,
the sections Brunsbüttel-Cuxhaven and Hamburg-Grauerort stored the greatest part of it in terms of mass (see supplementary
figure S-3). The ratio is the same: During the spring nitrogen was deposited in the sediment to be released in summer and
autumn, such that the annual cycle was nearly closed in January (Figs. 12c, S-3). During summer and early autumn, the
observed nitrate values in the nitrate salinity gradient were lower than the modeled nitrate levels.

The rates of pelagic and benthic respiration, bottom upward ammonium fluxes and denitrification (Fig. 13) reveal the great
importance of the sedimentary layer. In summer, the pelagic respiration increased between the weir and the harbor region,
where a rather stationary maximum of oxygen consumption established between May and September 2012 (Fig. 13a, Hamburg
station marked by the white dashed line). The pelagic respiration demonstrated fortnightly fluctuations due to the spring-neap
cycle. A secondary maximum of the pelagic oxygen consumption arose at the seaward end of the considered section. In
between the harbor region and the mouth of the estuary respiration manifested four periods of elevated respiration that
approximately coincided with the timing as zooplankton blooms in the section downstream of Elbe-km 650 (Figs. 13a, 10c).
The summertime consumption of oxygen by the sedimentary layer had its maximum in the harbor region (Fig. 13b), too, where
oxygen consumption rates reached 25-30 % of the local pelagic respiration between June and September. The regions of high
bottom respiration in summer also demonstrated the largest ammonium fluxes (Fig. 13c), which emphasizes the importance of
the sedimentary system for ammonium cycling. Denitrification rates were small in the free-flowing part of Elbe estuary, as
seen from time series of vertically-integrated and horizontally-averaged denitrification rates in the section Zollenspieker-
Hamburg (red solid line in Fig. 13d, see Fig. 1 for section location).  Denitrification rates in the sedimentary layer in the same
section reached 30 mmol N m$^{-3}$ d$^{-1}$ during summer (dashed red line in Fig. 13d). In the harbor area denitrification rates could
be higher even in the water column (black solid line in Fig. 13d). In the harbor sediments denitrification peaked, with up to
200 mmol denitrified per square-meter per day during three events in summer 2012 (black dashed line in Fig. 13d representing
spatial average over the marked harbor area in the figure inset). These relatively high rates resulted in the local reduction of



nitrate (Fig. 12 e), they would however not overcome the trend of nitrate production in the main channel until Elbe-km 650 (Figs. 8a, b; 10d).

Meso-zooplankton had the capability to bridge from the limnic regime to the outer estuary and more oceanic regime (Fig. 10c). During several pulses during the year the grazers built up considerable concentration of ~ 20 mmol N m$^{-3}$ along the salinity front. These pulses were triggered by a couple of small phyto-plankton blooms, not limited by silicate (Figs. 12d, 10b, c). In the lower reaches of the estuary, autotrophy and heterotrophy reached a balance as illustrated by the predator-prey oscillations near Cuxhaven (Fig. 12 d), de-coupling from the limnic regime described in the paragraphs above. The grazers produced

particulate nitrogen (Fig. 12d, 10c, h) which subsequently was remineralised to ammonium. Apparently respiration was high enough to locally reduce oxygen levels (Figs. 12d, 10d, i). In the poorly oxygenized water ammonium revealed a second estuarine maximum (around km 750 in Fig. 10e). Biomass levels in the outer estuary were higher than in the area of the nitrate maximum but still one order smaller than in the shallow limnic part of the estuary (Fig. 10a-c).

From the above description of the estuarine ecosystem dynamics during 2012 the following spatial three-fold partitioning of the estuary could be derived: 1) The limnic shallow stretch upstream from Hamburg (km 625, Fig. 1) with high concentrations of organic matter, 2) a region of intense turnover between the shallow stretch and the nitrate maximum, including in particular the harbor area, as identified by growth of meso-zooplankton (Fig. 10c), production of particulate nitrogen (i.e. detritus or pellets, Fig. 10h), recycling of ammonium (Fig. 10e) and an intense summer depletion of oxygen (oxygen undersaturation

defined by the blue areas in Fig. 10i). A third zone (3) comprises the the salinity intrusion and is identified by the pulses of small phyto-plankton blooms during spring and summer (Fig. 10b) revealing notable primary production that is absent in 2).

**5.2 Inter-annual comparison 2012 vs. 2013**

In 2013 river forcing differed from 2012 in several aspects: Winter and spring discharge was slightly greater and June was marked by a flood event (Fig. 3c). Residence times were smaller both on average and in terms of extremes (Fig. 3c). Small

residence times and decreased dispersion would oppose efficient remineralisation of river-borne nitrogen and ventilation of the lower estuary by oceanic waters. On the other hand surface temperatures were higher in 2013 (Fig. 3b) and could balance the change in kinetics. Finally, the riverine inputs in 2013 contained smaller concentrations of diatoms and organic matter to be remineralised but larger concentrations of nitrate, especially in the first half of the year (Fig. 10a, g, d).

Accordingly, levels of ammonium upstream from the salinity intrusion were generally lower than in 2012, except a peak in May associated with a peak forcing of organic matter (Fig. 10e). The increased river run-off shifted the remineralisation of ammonium and nitrate further downstream (Fig. 10e, d). The on-set of the early summer flood event was marked by decreasing concentrations of diatoms and oxygen and a peak of silicate concentrations at the weir (Fig. 10a, i, f). The flood flushed the grazer population usually residing in the harbor area downstream, bringing river-borne diatoms and small phyto-plankton until

the tip of the salinity intrusion (Figs. 10c, a, b; 3a). In the harbor meso-zooplankton was strongly reduced in June and July 2013 compared to 2012, such that primary producers reached higher concentrations (Fig. 12b). This led to higher oxygen saturation which did not develop the summer low seen in the first year of model integration (Fig. 12b). Accordingly ammonium levels were smaller than in 2012, reaching almost zero in July 2013 (Fig. 12a). However nitrate concentration peaked in the same time indicating that remineralised nitrogen was nitrified inside the basin owing to the high oxygen levels (Fig. 12a, b).


The shift of the heterotrophic chain due to the increased run-off was further reflected by reduced sedimentation rates upstream from Hamburg but an increase in the seaward direction (Fig. 12c). In the harbor itself sedimentation was also smaller. The high summer discharge perturbed the spatial pattern of ammonium remineralisation seen in 2012 with its maximum migrating upstream in the course of the year from Brunsbüttel (March) to Grauerort (April, May) to Hamburg (June-September) (Fig.

10e).

Particulate nitrogen peaked in summer 2013, revealing high productivity of the estuary during the increased river discharges. Nitrate behaved very differently to ammonium and particulate nitrogen because of its largely conservative mixing (Fig. 8). In the outer estuary high discharges were associated with elevated concentrations of grazers (Figs. 10c, 12d). They produced

more detritus and oxygen levels decreased (Fig. 12d). Oxygen depletion in the area of the German Bight in the aftermath of flood was also reported by other authors evaluating observations (Voynova et al., 2017). Our simulations supported their finding that the flood brought large amount of particulate matter into the German Bight as can be seen from the elevated grazer and near-bottom particulate nitrogen concentrations (Fig. 10c, h). In late summer, the estuary came back to its 'normal' state showing along-channel distributions of the biogeochemical state variables mostly similar to the 2012 case. This comparison

suggests that during a major flood event the estuary 1) looses its three-fold structure, 2) is perturbed in the spatio-temporal organization of remineralisation, 3) after the event quickly re-established its previous functioning (except the benthic part, Figs. 12c, supplementary S-3).

## 6 Conclusions

We used observations and simulations with a 3D coupled physical-biogeochemical model to quantify nitrogen transport and

turnover in the Elbe estuary, focusing on the region between the tidal weir and Cuxhaven. The simulations underline the function of the estuary as a transformation space of organic nitrogen to nitrate. Nitrate-salinity relationship generated from both measurements and model results revealed conservative mixing of nitrate at salinities greater than 10 psu and non-conservative mixing at low salinities during the biologically active seasons. The model also reproduced the spatio-temporal variability of other observed nutrients, ammonium and silicate, as well as oxygen saturation. Based on the successful validation

of the estuarine nutrient and oxygen dynamics we compared the biogeochemical estuarine dynamics during the period 2012-2013. The first year of integration demonstrated known from the literature stages of nitrogen cycling in the Elbe estuary: In



spring and summer the estuarine nitrogen cycle was dominated by diatoms' inputs, as well as inputs of labile organic nitrogen. The diatoms triggered growth of grazers in the harbor area. A dedicated analysis of the dynamics in the harbor area revealed the strongest sedimentation, near-bottom oxygen depletion, denitrification and release of ammonium. Simulation of currents

supported the connectivity between the harbor basin and the main channel implied by biology. The connectivity was physically based on vertical shear, with top currents directed on average into the basin and near-bottom currents vice-versa. The stratification of biological state variables within in the basin and the lateral exchange demonstrated clearly the necessity for coupled high-resolution 3D-modelling in order to tackle the heterotrophic decay in the harbor area.

Our hindcast of the estuarine dynamics in 2012 showed good agreement with previous studies, both observational and model-based. In this first year estuarine biogeochemistry revealed a three-fold structure with 1) dominance of primary producers in the shallow limnic part, 2) heterotrophic decay and remineralisation in the deeper limnic part (including in particular the harbor) and 3) predator-prey oscillations in the saline part. Despite this persistent structure, the seasonal migration of the ammonium maximum showed systematic modulation of the remineralisation process. In 2013 the structuring and rhythm of

summer nitrogen cycling was disturbed by a major river flood. Inorganic nutrients, plankton and poorly oxygenized water were pushed from the limnic part towards the outer estuary. The harbor area was annihilated as dominant recycling area such that high grazer concentrations, sedimentation, remineralisation and oxygen depletion were enhanced in the lower reaches. By September 2013 simulated Elbe estuary biogeochemistry manifested similar patterns and dynamics as in the previous year. Time series of the sediment budged demonstrated that in 2012 sedimentation and deposition were well balanced. Such could

not be expected in 2013 given the extreme river discharges.

The simulations proved that the nitrogen cycling in the Elbe estuary can be tackled with a robust NPZD model coupled to a high-resolution 3D hydrodynamical model. Simulations revealed high relevance of the harbor for heterotrophic degradation. Further observational studies seem necessary to support the proposed by the model dynamics in the harbor. Although the

presented herein approach worked convincingly, model development should focus on a more explicit formulation of the heterotrophic decay such as implementation of labile and refractory organic nitrogen.

**Acknowledgements**

We thank Michael Bergemann and the FGG Elbe for providing the observational data. Benjamin Jacob is acknowledged for his support with the set-up of the hydrodynamical forcing. This study has been partially funded by the Helmholtz Society's

HGF Excellence cluster for the DFG Excellence Cluster CliCCS (funding sign Ex-Net-0025).





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

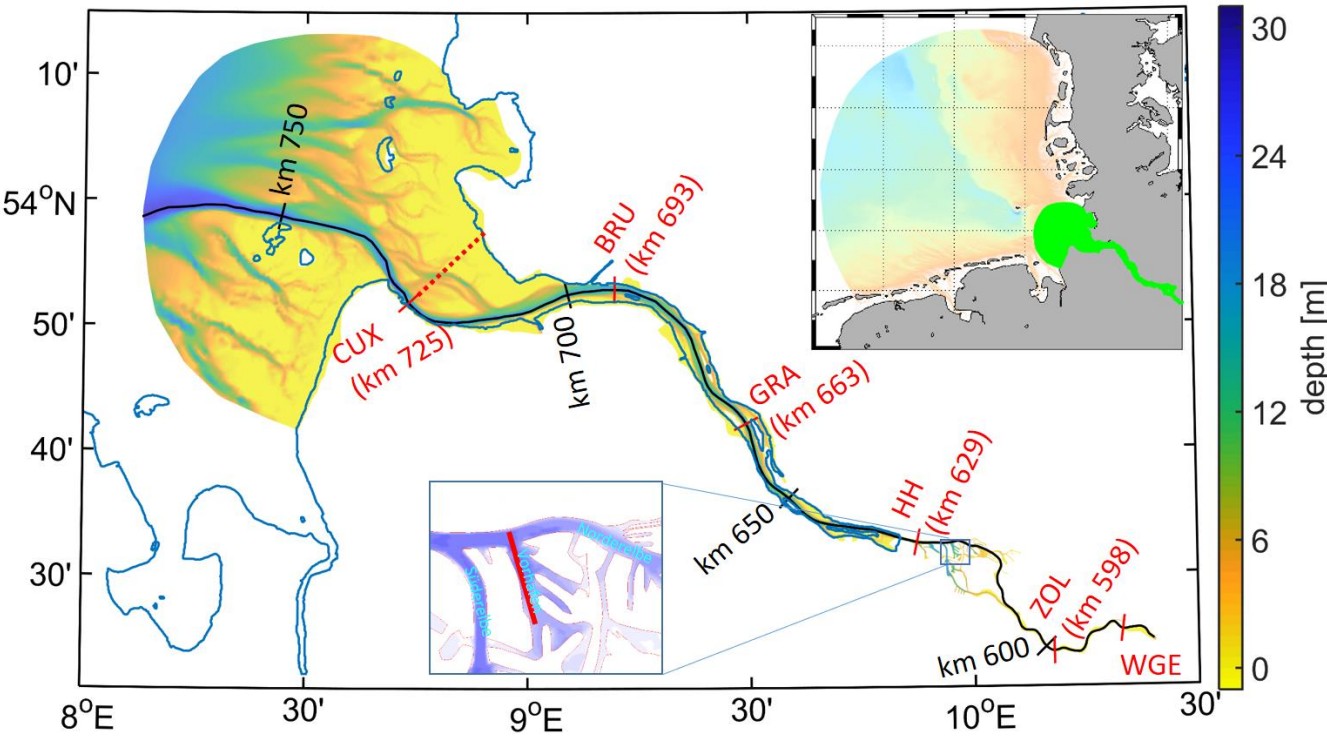

**Figure 1:** The model domain with depth [m] given as a background. The black transect line indicates the Elbe navigational channel with labels identifying the official Elbe-km. Red labels and associated ticks mark the location of observation stations with respective official Elbe-km given as additional information. A zoomed-in view illustrates the central part of the Hamburg harbor area with red solid line marking transect used for dynamical analysis. The inset at the top right marks the location of the model area (green color) within the German Bight set-up used to force the Elbe-model hydrodynamically.

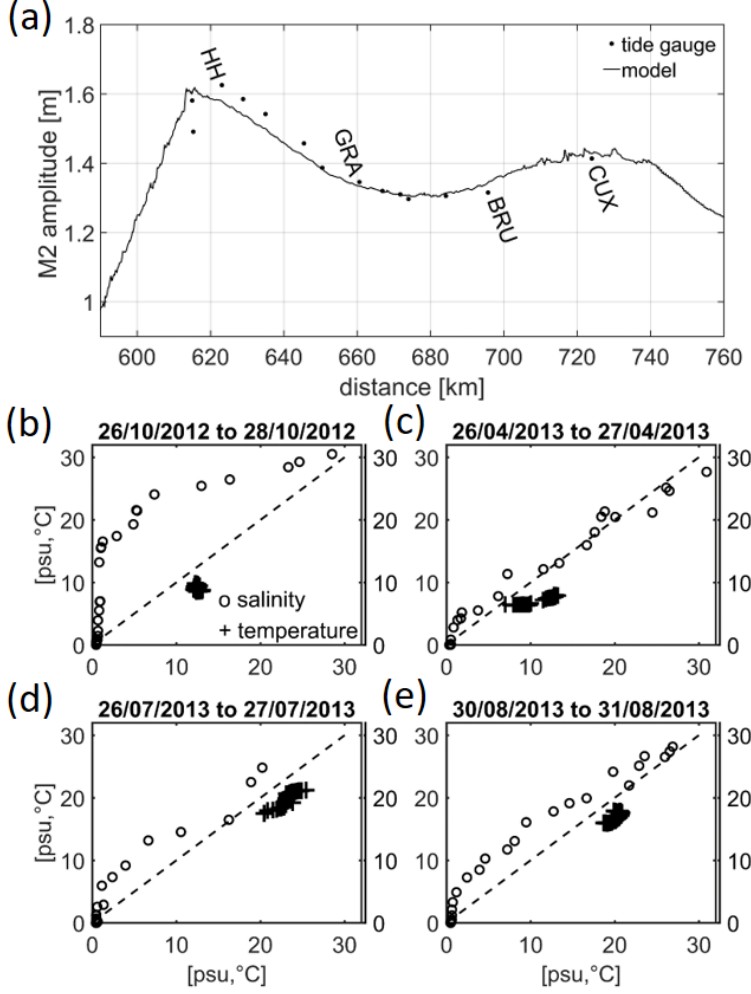

**Figure 2:** (a) The M-2 amplitudes of model elevation [m] along Elbe estuary (solid line) are compared with M-2 amplitudes derived from gauge measurents (black dots). In (b-e) black circs show model results of salinity versus salinity observations [psu] from ship-borne CTD

casts along Elbe estuary during 08/01/2012-29/09/2013. Black crosses represent model temperature [°C] versus observed temperature [°C] where axis scale of temperature equals axis scale of salinity.





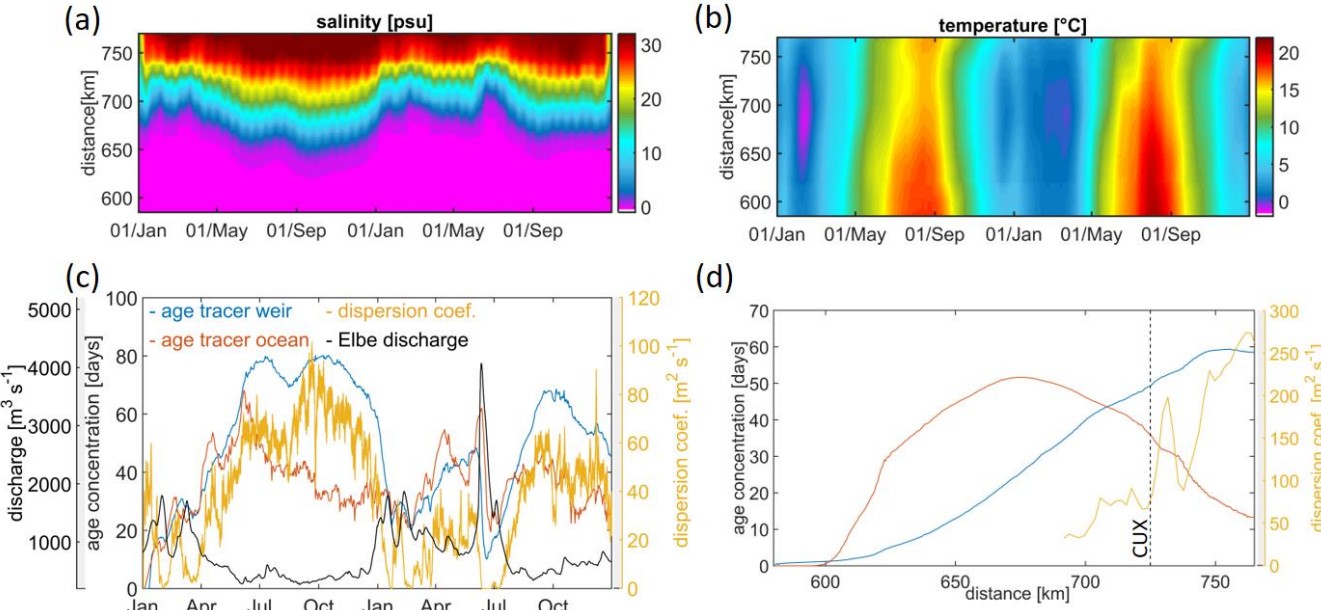

**Figure 3:** Time-vs.-distance diagrams of (a) sea surface salinity [psu] and (b) sea surface temperature [°C] along Elbe estuary (see Fig. 1 for

referencing km-scale to map positions). In (c) blue and red solid lines refer to the tracer age concentration at Cuxhaven (Elbe-km 725) of

age tracers injected into the model domain at Geesthacht and the open ocean boundary, respectively. The red solid line represents estimated

dispersion coefficient [m^2/s] at the same location while solid black line gives the Elbe freshwater discharge measured at Neu Darchau. (d)

illustrates the average along-channel distribution of tracer age concentration (blue line: tracer introduced at the tidal weir, red line: tracer

introduced at the open ocean boundary) as well as along-channel dispersion in the area of the salinity front. The vertical black dashed line

with label "CUX" indicates the position of Cuxhaven station referring to the along-channel coordinate.





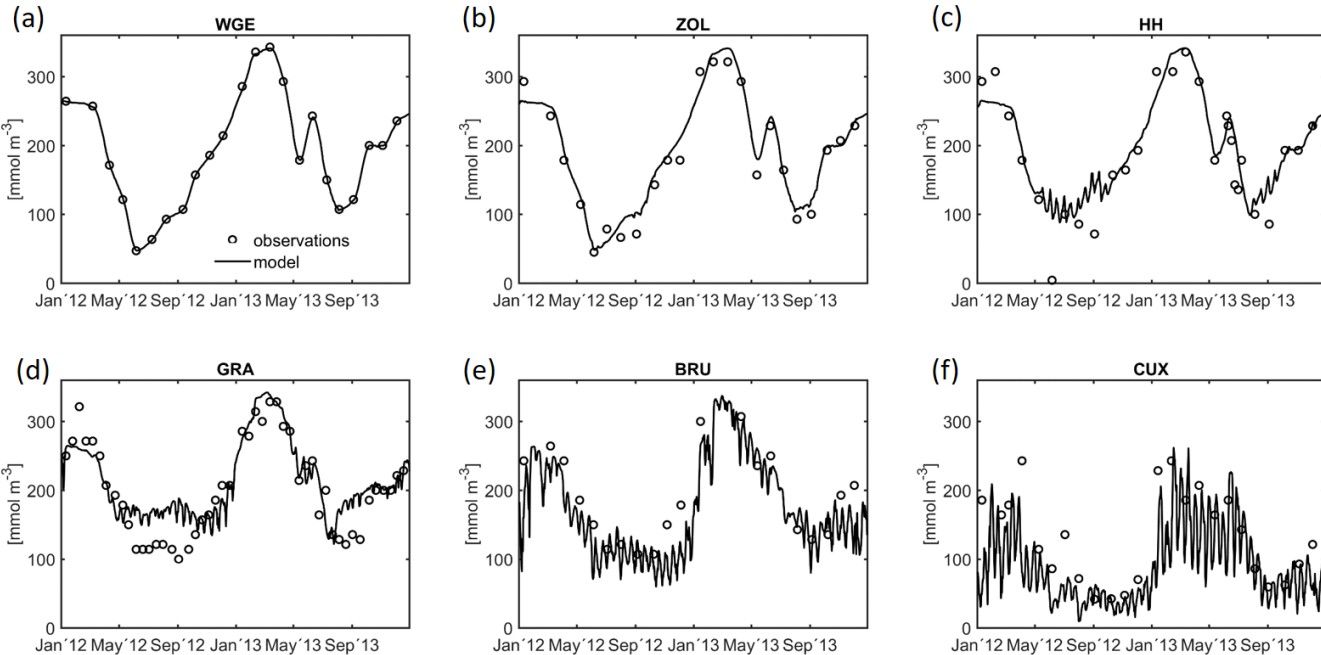

**Figure 4:** The observed surface nitrate concentrations at stations (a) weir Geesthacht -WGE, (b) Zollenspieker - ZOL, (c) Hamburg harbour - HH, (d) Grauerort - GRA, (e) Brunsbüttel - BRU, (f) Cuxhaven - CUX  are illustrated by the black circles. Solid lines represent the model time series of surface nitrate concentrations at the station positions.






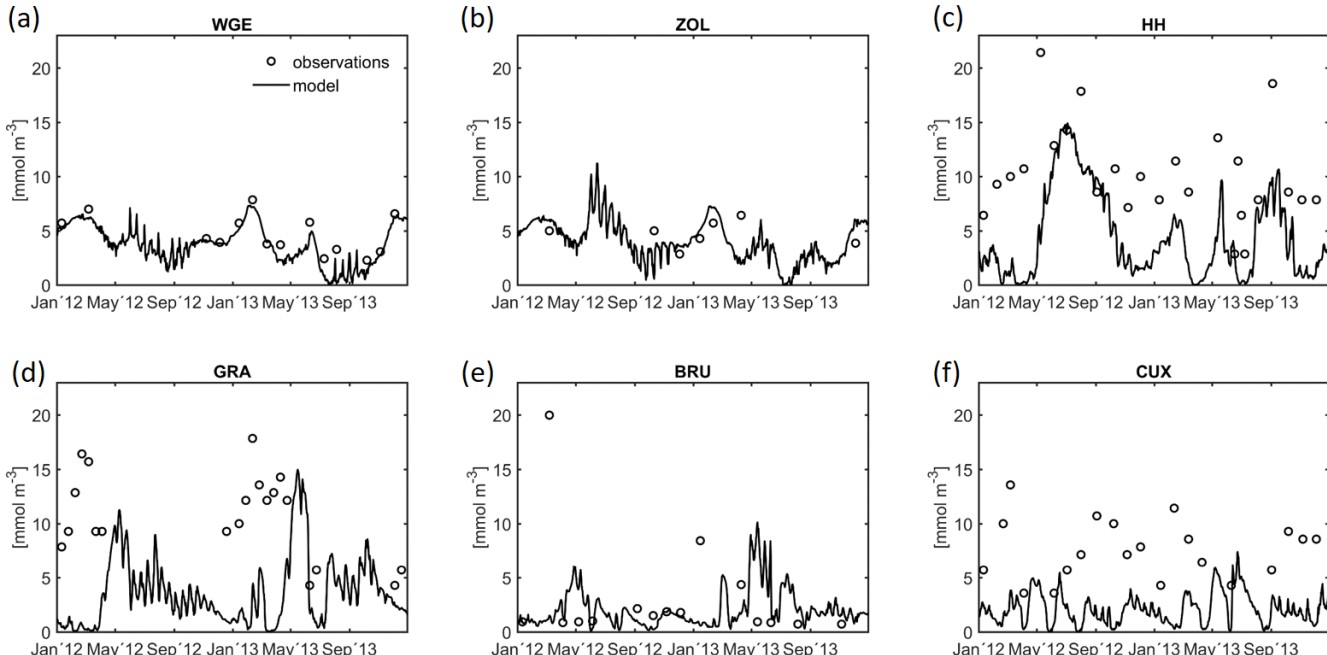

**Figure 5:** The observed surface ammonium concentrations at stations (a) weir Geesthacht -WGE, (b) Zollenspieker - ZOL, (c) Hamburg harbour - HH, (d) Grauerort - GRA, (e) Brunsbüttel - BRU, (f) Cuxhaven - CUX are illustrated by the black circles. Solid lines represent
the model time series of surface ammonium concentrations at the station positions.





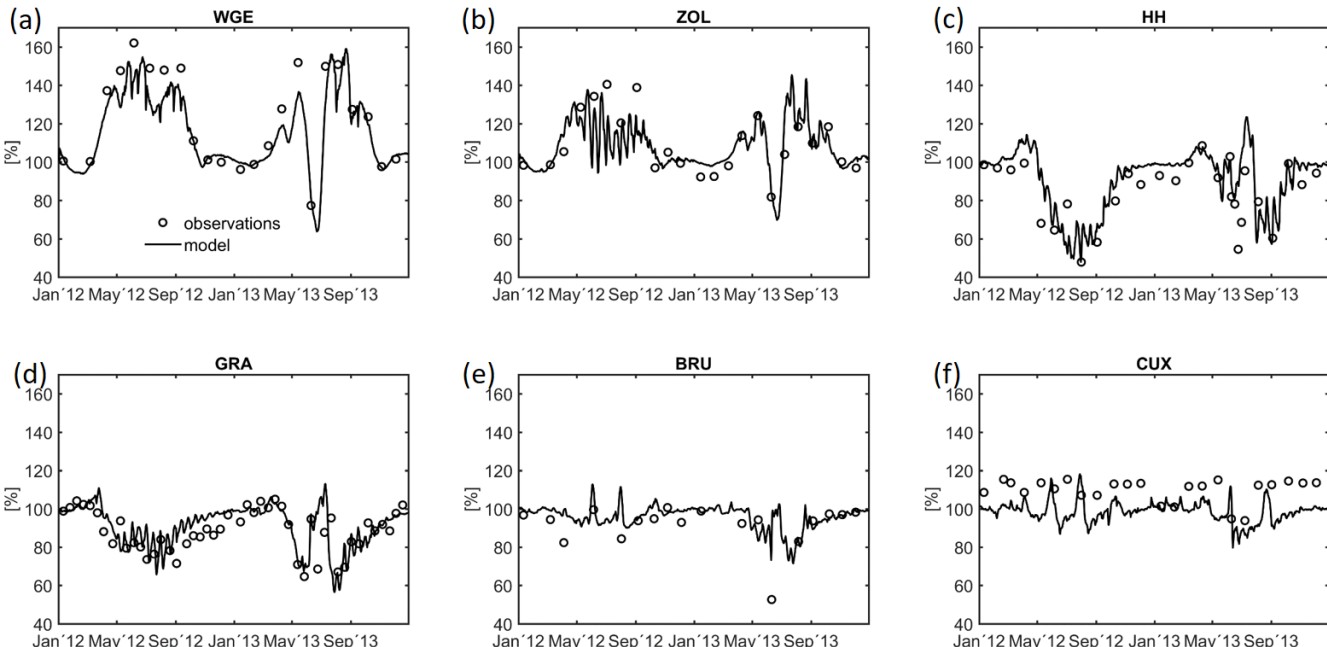

**Figure 6:** The observed surface oxygen saturation concentrations at stations (a) weir Geesthacht -WGE, (b) Zollenspieker - ZOL, (c)
Hamburg harbour - HH, (d) Grauerort - GRA, (e) Brunsbüttel - BRU, (f) Cuxhaven - CUX  are illustrated by the black circles. Solid lines
represent the model time series of surface oxygen concentrations at the station positions.





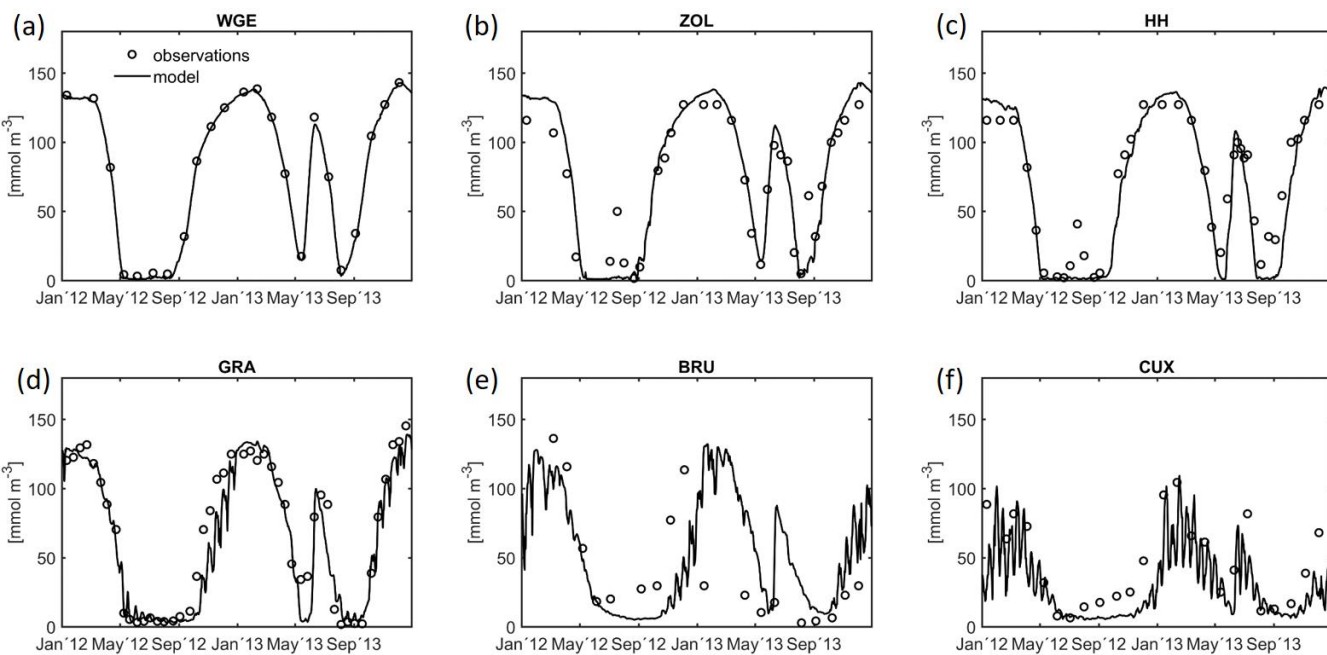


**Figure 7:** The observed surface silicate concentrations at stations (a) weir Geesthacht -WGE, (b) Zollenspieker - ZOL, (c) Hamburg harbour - HH, (d) Grauerort - GRA, (e) Brunsbüttel - BRU, (f) Cuxhaven - CUX  are illustrated by the black circles. Solid lines represent the model time series of surface silicate concentrations at the station positions.


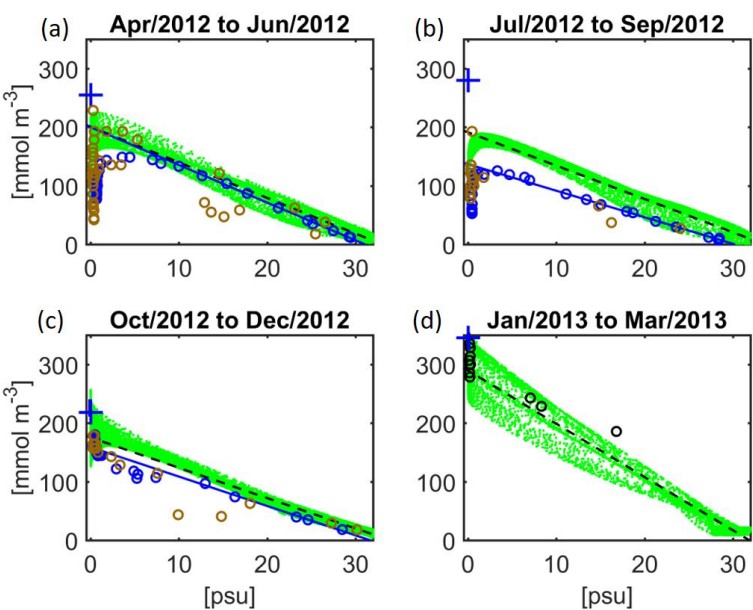

**Figure 8:** Nitrate-salinity relationship (mixing plot) as observed by ship (blue circles) and helicopter surveys (brown circles), whereas survey dates fell into the periods named on top of each panel (a-d). Green dots illustrate the same relationship as simulated by the model during the period named on top of each panel (a-d). Regression lines extrapolate the nitrate-salinity relationship from salinities $S > 10$ psu to salinities $S < 10$ psu, where blue solid lines and black dashed lines refer to ship-borne measurements and model results, respectively. The blue crosses at 0 psu indicate the magnitude of total nitrogen inputs at the tidal weir. Black circles in (d) represent the nitrate-salinity relationship as observed by administrative stations (FGG).





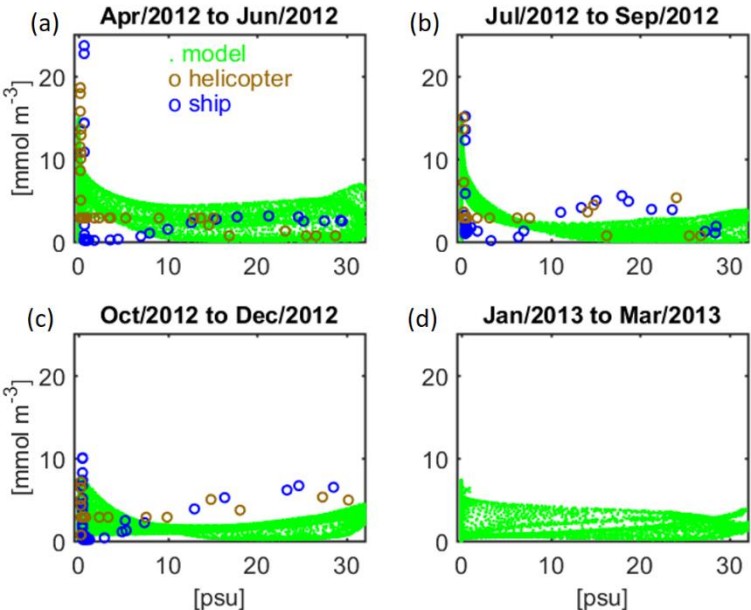

**Figure 9:** Ammonium-salinity relationship (mixing plot) as observed by ship (blue cicles) and helicopter surveys (brown circles), whereas survey dates fell into the periods named on top of each panel (a-d). Green dots illustrate the same relationship as simulated by the model during the period named on top of each panel (a-d).







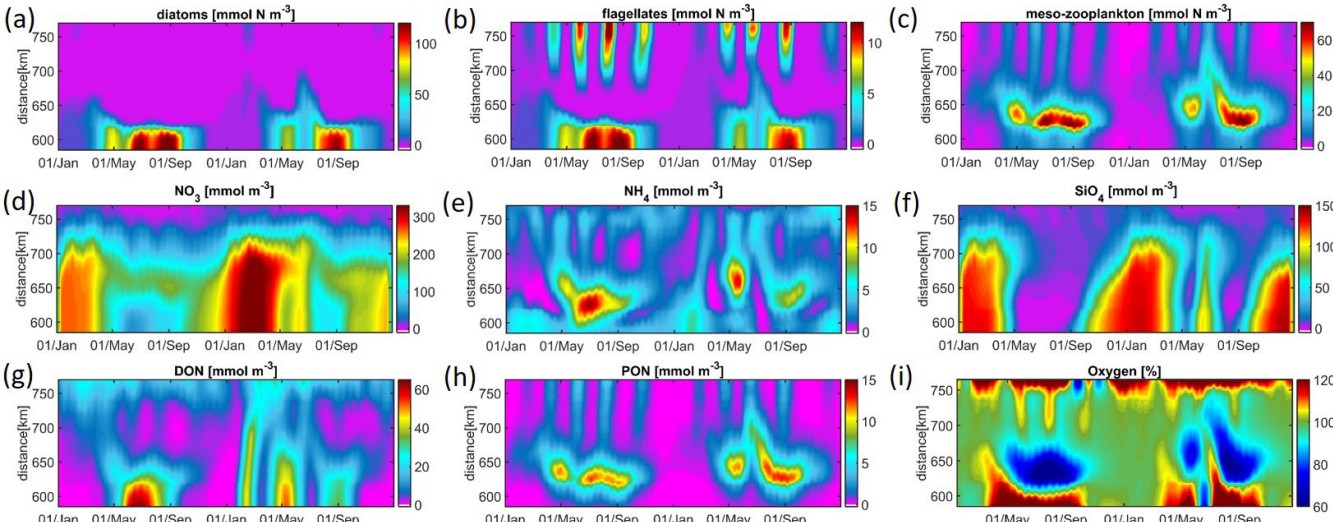

**Figure 10:** The seasonal variability of surface concentrations of (a) diatoms, (b) flagellates, (c) meso-zooplankton, (d) nitrate, (e) ammonium, (f) silicate, (g) dissolved organic matter, and (i) oxygen saturation along the Elbe estuary. (h) gives the near-bottom concentration of particulate organic nitrogen (detrite) where the white dashed line marks the Elbe-km 629 where the station Hamburg (see Fig. 1) is located. For better readability, the high-frequency variability has been removed from the data by applying a Gaussian filter with a time window of eight M2 periods.

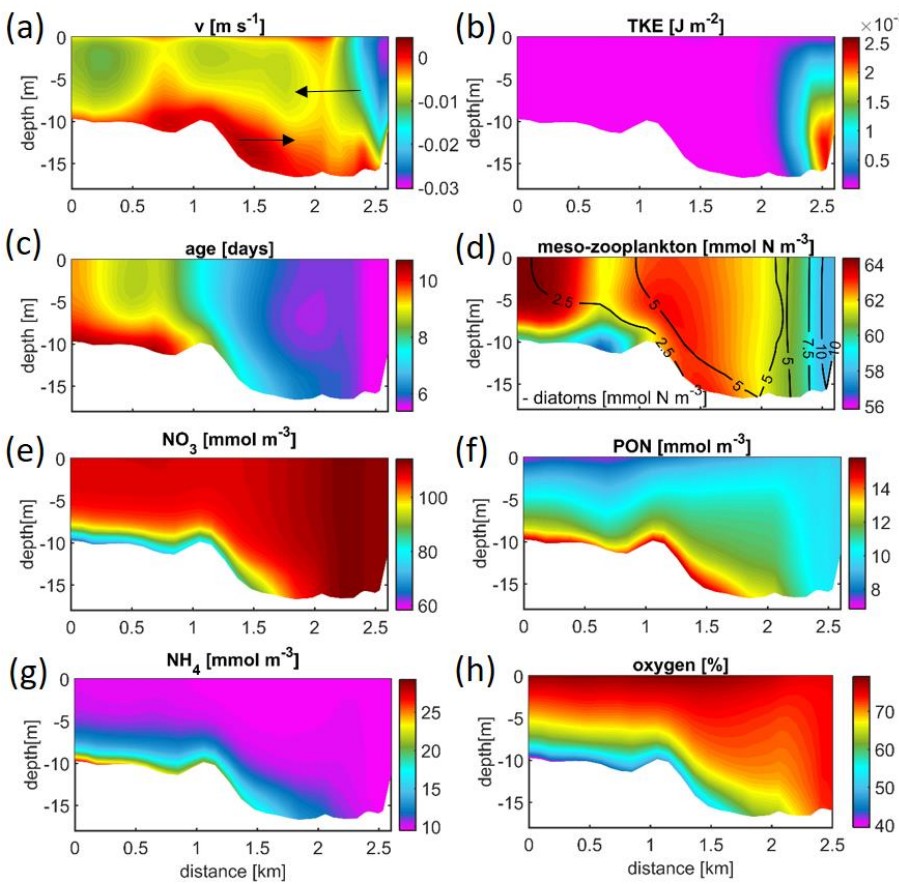

**Figure 11:** Averaged over the period of 1 May 2012 to 30 September 2012 along-transect profiles of (a) northern velocities where arrows illustrate along-transect circulation cell, (b) TKE, (c) mean tracer age concentration of a river-borne tracer, (d) meso-zooplankton (in mmol N m$^{-3}$) with contour lines representing concentration (in mmol N/m^3) of diatoms, (e) nitrate, (f) particulate organic nitrogen, (g) ammonium and (h) oxygen saturation along a transect through a harbour basin (*Vorhafen*) and across Elbe river (*Norderelbe*), for transect location see Fig. 1.



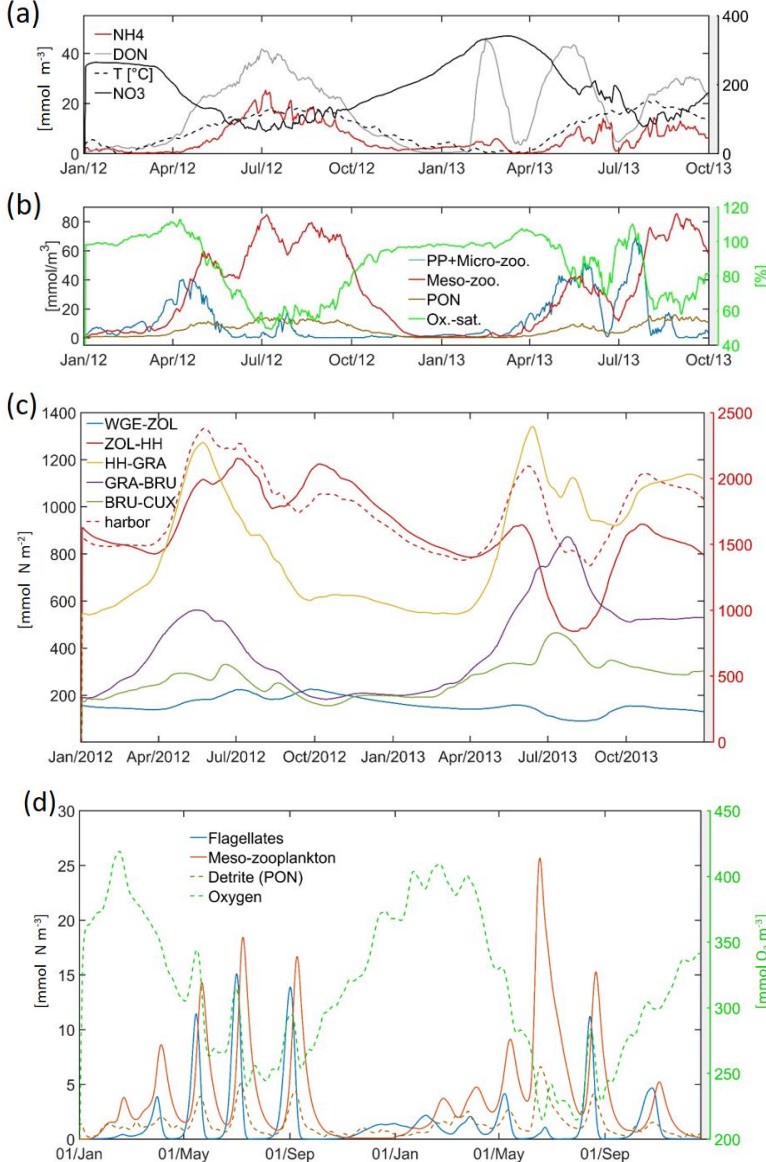

**Figure 12:** (a,b) time series of vertically averaged (a) ammonium concentration, concentration of dissolved organic nitrogen, water
temperature, nitrate concentration and (b) concentrations of diatoms, meso-zooplankton, particulare organic nitrogen and oxygen at the
southern end of a transect trough one major habor basin (see Fig. 1). (c) time series of sedimentation of detrite in different sections during
time of model integration (sections defined by areas in between stations shown in Fig. 1). Dashed red line gives the average sediment content
in one harbor basin, *Vorhafen* (see Fig. 1). (d) time series of simulated surface concentrations of flagellates (solid blue line), meso-
zooplankton (solid red line), detritus (dashed brown line) and oxygen (dashed green line) in the Elbe river near Cuxhaven station (see Fig. 1
for its position).





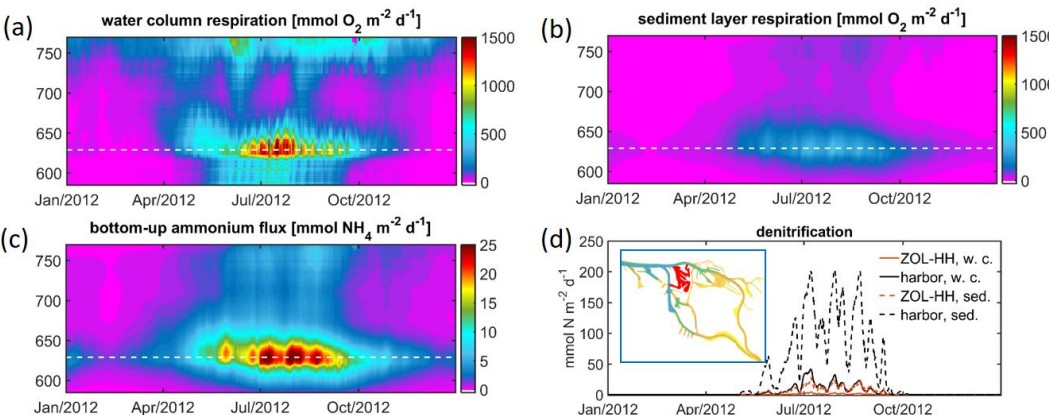

**Figure 13:** time-versus-distance diagrams of the local daily rates of (a) oxygen consumption by respiration in the water column, (b) oxygen consumption by respiration in the sedimentary layer, and (c) ammonium flux from the sedimentary layer to the water column. (d) gives the time-series of the spatially averaged rates of denitrification in the river section Zollenspieker-Hamburg (see legend "ZOL-HH") and in the harbor section marked red in the inset in (d) (see legend "harbor"), respectively, where solid lines represent denitrification in the water column (see legend "w.c.") and dashed lines represent denitrification in the sedimentary layer (see legend "sed.").