# Peer review of "Nitrogen cycling in the Elbe estuary from a joint 3D-modelling and observational perspective"

_Biogeosciences, 2019_

## Referee Comment (RC1) · Ingrid Holzwarth (Referee) · 19 Aug 2019

General comments

My overall impression is that the manuscript is a draft with a collection of several interesting topics (importance of a 3D model to capture estuarine biogeochemistry, nitrogen cycling, hydro-dynamic transport times, grazing, influence of river discharge, small-scale hydrodynamic pat-terns influencing the system-wide biogeochemistry) but no consistent story line and no clearly expressed conclusion in the end. My general advice is to focus on one aspect from the collection of interesting aspects above and present it in a clear and concise way.

The manuscript presents a work very specifically done for the Elbe Estuary without

discussing the global situation or the situation in other similar, or different, estuarine systems. Almost no literature apart from specific Elbe Estuary literature is included.

Though I agree on the necessity to use a 3D model for analyzing estuarine biogeo-chemistry, this point is not worked out well: (1) The argumentation for the necessity of a 3D model fails. (2) The authors make little use of the additional information they get from their 3D model compared to information from measurements or a 1D/2D model.

Apart from the issues above, the model (physical and biogeochemical) is not sufficiently validat-ed for the argumentation later on. While I see nitrogen and nitrogen species well-validated, all other quantities are not. In any case, the physical model need better validation, and – given the poor agreement shown for salinity – probably some more effort in setting up a better physical model is required.

Specific comments Title It is not clear what the authors mean with "joint perspective". The manuscript presents a model-ling study with the use of observational data for model steering and validation, which is the usual way to do.

Abstract Line 7-8: observations are only used for steering & validation here. Line 9: "a coupled physical-biogeochemical 3D unstructured model, applied for the first time to the estuarine environment". It is not clear, what the authors mean: if the sentence relates to the specific model they use (Schism – ECOSMO) it might be true but is nothing new and remarkable in the abstract. However, the expression could also be misleading: I'm not sure if the authors mean that it was the first time ever that a coupled model has been applied to an estuary. In this case, the authors are referred to Cerco et al. (1993), Wool et al. (2003), Wild-Allen & An-drewartha (2016), Lajaunie-Salla et al. (2017) – to name only some. Line 13 – 21: - Apart from the validation for nitrogen, model validation for all other physical and biogeo-chemical quantities is not sufficient to support these findings. - What is the new& important finding and what are its consequences?

Page 2 Line 31: The reference Jickells et al. 2014 is not appropriate for the statement.

Line 38-39: If oxygen depletion is mentioned later on, it should be mentioned that depletion was most severe during this period. Line 61-62: Sentence unclear: in x-direction (main flow direction), all of the above mentioned models describe the gradient between the weir and the German Bight. Entire last paragraph: the models mentioned before all focus on oxygen, the authors focus on nitrogen (or meso-zooplankton?). The authors should consider that this might be a completely different story.

Page 3 Line 70-74: limiting to one or two questions would probably help to give a more concise and focused article Line 75-81: the stations of WGMN Hamburg (Blankenese, Seemannshöft, Bunthaus) should also be taken into account. They provide high-quality data in an area where later on important con-clusions are drawn from model results. And they give a differentiation of phytoplankton between diatoms, green algae and blue-green algae. Line 80: Use boundary "values". At this point I miss the information about the location of the model boundaries. Line 85: it should be mentioned that measurements are taken approx. 1m below the water sur-face. No vertical (and no lateral) gradients available. Line 92-95: references missing

Page 4 Line 96-97: unclear: were climatologic averages used? Daily values? Is the same method used for all nutrient species? Line 108: What is GCOAST? Is it important information?

Page 5 Line 128: What exactly are the state variables? What is their functional relation? The authors say that organic matter decomposition was very important (which is true), so was the ECOSMO2 setting changed from the references given before? If it is not changed, this information should be clearly given here. In case it is changed (also in case certain parameters are changed), the model functions and their parameter values have to be described in more detail. Line 129: Assumes Redfield ratio in phytoplankton, I guess? Line 133: What kind of feedback is meant here? Line 134: Unclear: first, the authors say that the sediment is a mere storage. Then they say, min-eralization and nitrogen release was possible. Line 140-141: Holzwarth and Wirtz 2018 say: turbidity is so high that depth plays a secondary role. What is the value for the background

attenuation coefficient used here? In case it is the same value used for the North Sea it will certainly be too low. See comment above: biogeochemical model description is missing. Line 156: Which station exactly? What is the temporal resolution of the observational data? Line 159: Is the conversion factor in line with the conversion used in other Elbe studies (some of them were already cited)? If not, why different values were used? Entire paragraph with Geesthacht boundary values: please provide a plot containing the time series of the used boundary values.

Page 6 Line 164: Please clarify: 2012 was used as spin-up period? Line 171: Why can the authors assume, based on M2 tide, that water levels deviate less than 10 %? Besides: - I consider 10 % as far too poor, given a water depth more than 15 m in large parts of the estuary. This would imply a substantial error in water volume. - More information on tidal analysis for observational and modeled data (mean high and low water, tidal range, slack water times) would be helpful to assess the quality of the physical model, essentially including an analysis of current velocities. Line172: "ventilation" might be misleading in this context (could be understood as "reaeration") Line 174-175: The stationary monitoring data of the water and shipping administration, which are distributed along the salinity gradient, from Hamburg to Cuxhaven should be used for valida-tion. Using these data, a time series comparison for the entire simulation period, including basic statistics, is possible. This would give a much better impression on the quality of model results than comparing 4 single transects. Same for temperature. Line 176: The deviation in salinity is quite high. Given the really high overestimation in the Oc-tober 2012 profile (up to 15 psu, if I understand it correctly) I strongly recommend to include the stationary measurements, see above. Either the ship-based observational data are bad, or the physical model needs improvement. Line 181: I assume the authors refer to model results (please clarify, also in the figure text)? Giv-en the poor agreement between observational data and model results I recommend not to show this figure until better agreement is proven.

Page 7 Entire paragraph on dispersion and tracer: I recommend proving good performance of salt transport (see above) before calculating a dispersion coefficient based on salinity. More: the relevance of the entire section is not clear to me. I recommend leaving it out complete-ly and better spending more effort in the validation of the physical model. Besides, mixing plots for model data are unnecessary from my point of view: having a model, you can extract from your model results the production rates for all your state variables in your entire model domain. Then you clearly see (in numbers!) whether you have a biogeochemical activity or not. You can even see where it comes from. Maybe the authors can make better use of this information which is so far not used in their study.

Page 8 Line 232: where shown?

Page 9 Line 248-249: earlier you say that denitrification is included in the model. Line 271 ff: Regarding the rather poor agreement of model salinity and temperature, the oxygen values should not be validated using saturation. Dissolved oxygen concentration should be used instead to avoid any offset due to incorrect salinity and water temperature. Line 279: What means high accuracy? This comment relates to the entire validation section, both physical and biogeochemical: analysis of the comparison between observational data and model results seems to be done by plotting and "visual" judgement on the goodness of fit.

Page 10 I don't see an additional benefit for model validation from comparing mixing curves and I there-fore recommend leaving out this section.

Page 11 The entire chapter 5 has a structure which makes it hard to follow a central theme. Line 326: Announcing to explain the "full spacio-temporal dynamics of the estuarine ecosystem" is a very big promise and I strongly recommend being more moderate. Line 348-359: First, the model used in this study has not been validated for grazing or grazers. Second, the reference given (Schöl et al. 2014) is weak because the conclusion presented therein only bases on model results. Therefore, findings regarding grazers have to be validated, also to sustain the conclusions drawn from it

later on. Observational data (counts of organisms, for ex-ample) might exist at the WGMN Hamburg or at BfG for the harbor area. Moreover, the model has also not been validated for phytoplankton. High-quality phytoplankton data are available, especially for the area that shows high model diatom concentration in Figure 10 (a).

Page 12 Line 365-367: The authors don't prove this statement. Line 367 ff: The relation to the text in the first section of this paragraph isn't clear. Line 376 and following two sections: When focusing to such a small region compared to the en-tire model domain, the authors should provide better validation for the hydrodynamic model in this specific area. A system-wide M2-comparison, which also shows its highest deviations in the focus region (roughly km 620 in figure 2(a)), is not sufficient. In addition: lacking validation for phytoplankton and grazers, see above.

Page 13 Line 415 – Page 15 Line 476 This section can be shorted substantially be-cause it contains many well-known and well-described processes and relations (con-tained in the references of the paper). The three – zone - description in the last section (Line 470 – 476) has first been described by Schroeder (1997). New aspects worked out by the authors are not clear. Line 458: where is fig 12e?

Page 15 Line 478 – Page 16 Line 512 What is new &relevant in this section?

Page 16 Chapter Conclusions Line 514 –522: Everything that is mentioned here has been shown already several times, for the Elbe and for other estuaries. A 3D model that is able to simulate all these processes is also not new but more or less common (see the comment on the abstract). Line 523 – 528: Validation missing Line 530 – 540: This section is more a description. What is the important conclusion out of it? Line 542 – 543: This has already been shown for different estuarine systems worldwide. Line 543: From my point of view the importance of the harbor is the most interesting aspect that remains. However, I miss: 1. Better validation, 2. more focus, 3. analyzing hydrodynamic aspects as well (e.g. larger water volume in the harbor compared to section upstream), 4. comparison to other similar systems (like Guadalquivir-Sevilla,

Loire, Humber or other estuaries containing large ports).

Figures There are many figures. Some figures (like 8, 9 or especially 12) are not easy to understand. Some panels (especially Figure 13) are very small. Check for Figure 2(b) to (e) if the description of x and y axis and the conclusion that in the model salinity intrudes too far into the estuary is consistent. There is no white dashed line visible in Figure 10.

Technical correction There is a plenty of typing errors, careless mistakes and wrong word order, some obviously due to copy& paste. I'm not a native English speaker, and therefore I'm not in the position to give suggestions for language corrections. However, I have the impression that language, including grammar and use of uniform tenses, needs substantial improvement.

References

Cerco, C. F., and T. Cole (1993), Three-dimensional eutrophication model of Chesapeake Bay, Journal of Environmental Engineering, 119(6), 1006–1025, doi:10.1061/(ASCE)0733-9372(1993)119:6(1006).

Lajaunie-Salla, K., K. Wild-Allen, A. Sottolichio, B. Thouvenin, X. Litrico, and G. Abril (2017), Impact of urban effluents on summer hypoxia in the highly turbid Gironde Estuary, applying a 3D model coupling hydrodynamics, sediment transport and biogeochemical processes, Journal of Marine Systems, 174, 89–105, doi:10.1016/j.jmarsys.2017.05.009.

Schroeder, F. (1997), Water quality in the Elbe estuary: Significance of different processes for the oxygen deficit at Hamburg, Environmental Modeling and Assessment, 73–82

Wild-Allen, K., and J. Andrewartha (2016), Connectivity between estuaries influences nutrient transport, cycling and water quality, Marine Chemistry, 185, 12–26, doi:10.1016/j.marchem.2016.05.011.

Wool, T. A., S. R. Davie, and H. N. Rodriguez (2003), Development of three-dimensional hydrodynamic and water quality models to support total maximum daily load decision process for the Neuse River Estuary, North Carolina, Journal of Water Resources Planning and Management, 129(4), 295–306, doi:10.1061/(ASCE)0733-9496(2003)129:4(295).

---

## Referee Comment (RC2) · Anonymous Referee #2 · 2 Sep 2019

There are many major issues to this submission:

1) There is no/little hydrodynamic calibration. Authors need consider to submit two manuscripts: one for hydrodynamic and one for water quality dynamics.

2) Why authors didn't calibrate the water quality for the bottom part, particularly to the oxygen? Ammonia simulation is a little different from the observed one, any justification?

3) The model set up and data description is very weak, and need a lot of work to this part. Again, authors need consider to split this manuscript into two manuscripts. Why choose year 2012 and 2013?

3) This study is very local, and there is no linkage to broad area? What is the contribu-

tion of this work the research community? The questions is pretty local, and not novel? Authors even didn't fully answer the questions of introduction part.

4) The mixing diagram was used by Jiang and Xia, 2018 and isn't new. This study is mainly for nitrogen dynamics, however authors want to cover everything. It is a little bit difficult to follow, and authors need think how to make a nice flowchart to this manuscript. Overall, it reads like a modeling or technical report.

There are many minor issues, however I would like authors to take care of major issues now.

Jiang, L. & Xia, M. (2018), Modeling investigation of the nutrient and phytoplankton variability in the Chesapeake Bay outflow plume. Progress in Oceanography, 162, 290-302. doi: https://doi.org/10.1016/j.pocean.2018.03.004

---

## Author Comment (AC1) · 23 Sep 2019

**Answers to Referee 1**

We thank Ingrid Holzwarth for her dedicated effort to read and comment on our paper. In the following we will give our point-by-point answers to the individual comments or paragraphs of her review. Reviewer's comments are written in italics, while authors' answers are in kept in plain font.

*General comments My overall impression is that the manuscript is a draft with a collection of several interesting topics (importance of a 3D model to capture estuarine biogeochemistry, nitrogen cycling, hydro-dynamic transport times, grazing, influence of river discharge, small-scale hydrodynamic pat-terns influencing the system-wide biogeochemistry) but no consistent story line and no clearly expressed conclusion in the end. My general advice is to focus on one aspect from the collection of interesting aspects above and present it in a clear and concise way.*

We are glad that the reviewer discovers several interesting aspects to our work. We agree that the paper needs further streamlining and focusing on the most relevant aspects. At the same time we would like to remind here that a complex system such as the coupled physics and biogeochemistry of the Elbe estuary require an integrate description. An excess of fractioning the integrate system into sub-systems and aspects increases the danger of overlooking important interconnections. For example we could not explain a spatial shift in a nutrient pattern without accounting for advection/diffusion of salinity and freshwater discharge, nor could we do so without accounting for organic inputs at the weir or diatom growth in the shallow limnic part. Our work demonstrates that the specific spatio-temporal organization of the nitrogen turnover in Elbe estuary depends on the physical and biogeochemical processes around the bifurcation of the river near Hamburg and specifically in the shielded from the tidal currents harbor basins and side channels. Following the reviewer's advice, in the revised manuscript we will focus on how the specific dynamics in the Hamburg port area affect the nitrogen cycling in Elbe estuary.

*The manuscript presents a work very specifically done for the Elbe Estuary without discussing the global situation or the situation in other similar, or different, estuarine systems. Almost no literature apart from specific Elbe Estuary literature is included*

We agree that the linking between our present work and its relevance for other estuarine systems needs to be improved. We will provide a better representation of studies on other estuaries as well as a statement on the global relevance of our study in the revised version of the manuscript.

*Though I agree on the necessity to use a 3D model for analyzing estuarine biogeochemistry, this point is not worked out well: (1) The argumentation for the necessity of a 3D model fails.*

A 3D model is necessary in case axial gradients, lateral exchange processes and stratification of physical and biogeochemical properties interact and control one another. The coupled physics and biology in the main channel of Elbe river could and have been analyzed in a simplified manner using 1D or 2D representations of the river geometry. In the area of the Hamburg port however, stratification, lateral gradients and exchange feed back onto the longitudinal dynamics (as seen for example in both observations and model from the estuarine ammonium maximum, a sharp peak located about Elbe-km 625, Fig. S-2). That is why a 3D model is necessary to increase the realism of the simulation beyond what has been achieved in the past modelling. This knowledge might also be useful for estuarine management. We find

that the benefits put forth by the unstructured 3D coupled physical-biogeochemical modelling in this particular case are of general scientific interest and deserve to be made public.

*(2) The authors make little use of the additional information they get from their 3D model compared to information from measurements or a 1D/2D model.*

While aspects of the advantage of the highly resolved model have been partially addressed by our analysis (e.g. along-section profiles in Fig. 11) we agree that additional information retrieved specifically form a unstructured 3D model will improve our manuscript. Following the referee's advice we will add time-averaged maps of the state variables most relevant to the heterotrophic decay around the channel bifurcation and port area to the manuscript. Such maps clearly reveal the hot-spots of remineralisation and hypoxia in the small channels and basins.

*Apart from the issues above, the model (physical and biogeochemical) is not sufficiently validated for the argumentation later on. While I see nitrogen and nitrogen species well-validated, all other quantities are not. In any case, the physical model need better validation, and – given the poor agreement shown for salinity – probably some more effort in setting up a better physical model is required.*

The Elbe set-up is a derivative of an already published set-up (Stanev et al. 2019) and it is also hydrodynamically coupled to the latter (Page 4 Line 114-116). Therefore, the hydrodynamical model of the herein presented coupled physical-biogeochemical model has actually been validated for the Elbe region and underwent the peer review process. However, the period of integration has been extended and the simulation has been compared to new observations available, i.e. the ship-borne measurements. We admit that simulation of salinity for certain periods in 2012 does not match expectations. There can be several reasons such as for example non-optimal boundary forcing or overestimated horizontal diffusivity. Both, operational model development and control of physical diffusion are developing topics of the modelling community. We agree that additional validation is helpful and will increase credibility of the model. We will provide additional validation for salinity, temperature and entangled with the nitrogen turnover biogeochemical properties using the available stationary observations. We will deliver appropriate tables and diagrams to provide a decent overview about the physical and biogeochemical model performance for the period of model integration.

*Specific comments Title It is not clear what the authors mean with "joint perspective". The manuscript presents a model-ling study with the use of observational data for model steering and validation, which is the usual way to do.*

We agree that in particular the second part of the paper is essentially a modelling study. On the other hand we use observations for complementary analysis of nitrogen turnover processes (Figs. 8, 9). As a compromise we propose to change the manuscript title as "Nitrogen cycling in the Elbe estuary: Observations and numerical modelling".

*Abstract Line 7-8: observations are only used for steering & validation here.*

We agree that we use observations for boundary forcing and validation, however complementary analysis of estuarine nitrogen cycling is based on observations. For example

on page 10 Line 307-309 we have written "In autumn, observations revealed concave bending at low salinities, indicating a nitrate sink in this area (Fig. 8c) or gradually increasing nitrate concentrations in the river Elbe (Fig. 4a)."

*Line 9: "a coupled physical-biogeochemical 3D unstructured model, applied for the first time to the estuarine environment". It is not clear, what the authors mean: if the sentence relates to the specific model they use (Schism – ECOSMO) it might be true but is nothing new and remarkable in the abstract. However, the expression could also be misleading: I'm not sure if the authors mean that it was the first time ever that a coupled model has been applied to an estuary. In this case, the authors are referred to Cerco et al. (1993), Wool et al. (2003), Wild-Allen & An-drewartha (2016), Lajaunie-Salla et al. (2017) – to name only some.*

We agree that our formulation is not optimal. Our statement intended to stress two aspects: 1) we present the first unstructured coupled 3D modelling using realistic bathymetry of Elbe estuary, 2) for a first time the ECOSMO biogeochemical model is applied in the estuarine environment. We thank the reviewer for recommending above references. The cited studies are certainly relevant for our study as they employ 3D coupled physical-biogeochemical model in the estuarine context. In the context of discussing the novelty of our modelling effort, we would like to point out that non of these modelling studies used unstructured mesh to resolve small scale features and curved narrow channels. However we find useful information in particularly in the more recently published of the above cites studies and we might integrate their findings into our reasoning.

*Line 13 – 21: - Apart from the validation for nitrogen, model validation for all other physical and biogeo-chemical quantities is not sufficient to support these findings. - What is the new& important finding and what are its consequences?*

The new and important aspect of our work resides in the use of an unstructured mesh in the context of 3D coupled physical-biological modelling of an estuarine ecosystem using realistic topography. Such a tool enables the integration of coupled dynamics in narrow, curved channels and harbor basins of Elbe estuary. Our modelling study reveals the hot spots of heterotrophic decay of the Elbe estuary ecosystem. To our knowledge, the past modelling studies have not elaborated on this aspect of the estuarine ecosystem dynamics. Following the reviewer's advice, we will better clarify the novelty of our work and its consequences for the understanding of the ecosystem in the revised manucript. Depending on the availability of observational data, we will also deliver additional validation of the relevant for our reasoning variables.

*Page 2 Line 31: The reference Jickells et al. 2014 is not appropriate for the statement.*

We will look for an appropriate reference to or change the statement. A possible reference would be Weilbeer (2014), who mentions ongoing maintenance dredging which does however not imply morphological changes in the future.

*Line 38-39: If oxygen depletion is mentioned later on, it should be mentioned that depletion was most severe during this period.*

We will add this information to the manuscript.

*Line 61-62: Sentence unclear: in x direction (main flow direction), all of the above mentioned models describe the gradient between the weir and the German Bight.*

We will reformulate this phrase and point out that herein also the feedback of vertical and lateral gradients and exchange processes on to the along-channel gradient is taken into account.

*Entire last paragraph: the models mentioned before all focus on oxygen, the authors focus on nitrogen (or meso-zooplankton?). The authors should consider that this might be a completely different story.*

It is correct that the cited studies focus on oxygen. In our model, nitrogen and oxygen concentrations are dynamically linked through production and degradation processes. We will clarify the interdependency of nitrogen and oxygen in the revised manuscript. We will also add a statement about the specific approach implemented in ECOSMO in comparison with the previous studies on the Elbe system.

*Page 3 Line 70-74: limiting to one or two questions would probably help to give a more concise and focused article.*

We agree that the paper needs substantial streamlining. As said above, we see the novelty of this work in addressing degradation processes in the port area. We will re-organize the manuscript such that this main aspect is better exposed.

*Line 75-81: the stations of WGMN Hamburg (Blankenese, Seemannshöft, Bunthaus) should also be taken into account. They provide high-quality data in an area where later on important conclusions are drawn from model results. And they give a differentiation of phytoplankton between diatoms, green algae and blue-green algae.*

Some of this data are indeed available for the given period. Blankenese data is currently not available according to the online tool. In particular the chlorophyll-a data available for stations Bunthaus and Seemannshoeft- that is upstream and downstream from the major part of the port area- could be useful for validation and reasoning. Following the reviewer's advice, we will make use of these data.

*Line 80: Use boundary "values". At this point I miss the information about the location of the model boundaries.*

The landward model boundary is at Geesthacht (Fig. 1). The stationary measurements at the tidal weir at Geesthacht (Elbe-km 585.9) were used to force the model at its landward boundary. We have specified this on page 5 line 154-156 of the manuscript. We will remove the statement about model boundaries at line 80 to avoid confusion at this point of the manuscript.

*Line 85: it should be mentioned that measurements are taken approx. 1m below the water surface. No vertical (and no lateral) gradients available.*

We will add this information to the manuscript.

*Line 92-95: references missing*

We will add references.

*Page 4 Line 96-97: unclear: were climatologic averages used? Daily values? Is the same method used for all nutrient species?*

We used daily valus of river runoff (cf. line 91). Nutrient observations were sparser in time. We generated a daily time series of observed nutrients using cubic spline interpolation. We used the same method for all nutrient species. We will clarify that in the text.

*Line108: What is GCOAST? Is it important information?*

The "Geesthacht COAstal model SysTem" (GCOAST) refers to a high-resolution modelling tool that couples ocean, wave and atmosphere dynamics. It is currently under development at Helmholtz-Zentrum Geesthacht. This could be relevant information because the modelling of estuarine ecosystem dynamics presented in this study is part of a greater effort to develop a modelling tool for research into the entire land-ocean transition zone.

*Page 5 Line 128: What exactly are the state variables? What is their functional relation? The authors say that organic matter decomposition was very important (which is true), so was the ECOSMO2 setting changed from the references given before? If it is not changed, this information should be clearly given here. In case it is changed (also in case certain parameters are changed), the model functions and their parameter values have to be described in more detail.*

The biogeochemical model we use, ECOSMO2, is in detail described in Daewel and Schrum, 2013. The state variables and processes remained unchanged. The parametrization remained largely unchanged except we allow denitrification to happen at oxygen concentrations below 120 mmol m$^{-3}$(line 135-136 in manuscript). The interaction between the water column and the sedimentary layer remained unchanged as well, it has however been implemented into the FABM framework. We will add an illustration of the FABM module and a table of the used in this study parameters and coefficients to the revised manuscript

*Line129: Assumes Redfield ratio in phytoplankton, I guess?*

Yes right, as was written in line 130 of the manuscript.

*Line 133: What kind of feedback is meant here?*

We will remove this phrase from the manuscript.

*Line 134: Unclear: first, the authors say that the sediment is a mere storage. Then they say, mineralization and nitrogen release was possible.*

Yes, it is right that the sediment is a mere storage, i.e. a reservoir of nutrients at the bottom. This reservoir gets filled by sedimentation. It can be depleted by remineralisation and

resuspension of the nutrient. We will provide a better understandable description in the revised manuscript.

*Line 140-141: Holzwarth and Wirtz 2018 say: turbidity is so high that depth plays a secondary role. What is the value for the background attenuation coefficient used here? In case it is the same value used for the North Sea it will certainly be too low. See comment above: biogeochemical model description is missing.*

We will correct this statement. The background attenuation coefficient was 0.05 $1m^{-1}$. We will add a table with the coefficients used to the revised manuscript

*Line 156: Which station exactly? What is the temporal resolution of the observational data?*

We refer above line 91-92 to daily discharge measured at Neu Darchau. Will be clarified in the revised manuscript.

*Line 159: Is the conversion factor in line with the conversion used in other Elbe studies (some of them were already cited)? If not, why different values were used? Entire paragraph with Geesthacht boundary values: please provide a plot containing the time series of the used boundary values.*

Hillebrand et al. (2018) give estimates of the Chl(a)-to-carbon ratio in Elbe river ranging from 18.2 µg Chl-a/mg C during winter months up to 32,3 µg Chl-a/mg C during summer months. Herein cited studies used values between ~27 µg Chl-a/mg C and ~ 50 µg Chl-a/mg C (Schöl et al., 2014; Holzwarth and Wirtz, 2018). Given that in our model C:N ratio equals 106:16 we effectively used a conversion value of ~ 20 µg Chl-a/mg C. We used this value following the applied by Zhou et al. (2017) 1,58 g Chl-a/ mol N. Assuming Redfield ratio of C:N this value appears to be in the range of Chl-a/mg C proposed by Hillebrand et al. (2018) for Elbe river.

Following the reviewer's advice, we will provide time series of the used boundary values at the tidal weir in the revised version.

*Page 6 Line 164: Please clarify: 2012 was used as spin-up period?*

Yes, 2012 was ran as spin-up and then repeated as actual herein presented simulation. We will clarify this in the revised manuscript.

*Line 171: Why can the authors assume, based on M2 tide, that water levels deviate less than 10 %? Besides: - I consider 10 % as far too poor, given a water depth more than 15 m in large parts of the estuary. This would imply a substantial error in water volume. - More information on tidal analysis for observational and modeled data (mean high and low water, tidal range, slack water times) would be helpful to assess the quality of the physical model, essentially including an analysis of current velocities.*

We agree that based on assessment of model performance regarding the M2-tide only it is unlikely to assure that errors are less than 10%. Model performance is actually better (see also Stanev et al., 2019). We will add a quantitative skill assessment for the period of model integration (2012-2013) for available along Elbe estuary gauge data in the revised version.

*Line172: "ventilation" might be misleading in this context (could be understood as "reaeration")*

By ventilation we mean forcing and stirring of the estuary by tides. We agree it could be misunderstand in terms of reaeration. We will reformulate this phrase.

*Line 174-175: The stationary monitoring data of the water and shipping administration, which are distributed along the salinity gradient, from Hamburg to Cuxhaven should be used for validation. Using these data, a time series comparison for the entire simulation period, including basic statistics, is possible. This would give a much better impression on the quality of model results than comparing 4 single transects. Same for temperature.*

We agree that stationary measurements of salinity and temperature should be taken into account. We will provide corresponding plots and basic statistics in the revised version of the manuscript.

*Line 176: The deviation in salinity is quite high. Given the really high overestimation in the October 2012 profile (up to 15 psu, if I understand it correctly) I strongly recommend to include the stationary measurements, see above. Either the ship-based observational data are bad, or the physical model needs improvement.*

To better address this comment we will provide comparison of model vs. observations using stationary data in the revised version of the manuscript.

*Line 181: I assume the authors refer to model results (please clarify, also in the figure text)? Given the poor agreement between observational data and model results I recommend not to show this figure until better agreement is proven.*

Yes, here we refer to seasonal variation of simulated salinity. We might change Fig. 3a for a stationary time series showing seasonal variability of salinity in both model and observations.

*Page 7 Entire paragraph on dispersion and tracer: I recommend proving good performance of salt transport (see above) before calculating a dispersion coefficient based on salinity. More: the relevance of the entire section is not clear to me. I recommend leaving it out completely and better spending more effort in the validation of the physical model. Besides, mixing plots for model data are unnecessary from my point of view: having a model, you can extract from your model results the production rates for all your state variables in your entire model domain. Then you clearly see (in numbers!) whether you have a biogeochemical activity or not. You can even see where it comes from. Maybe the authors can make better use of this information which is so far not used in their study.*

The model-derived dispersion coefficient illustrates the seasonal modulation of along-channel dispersion. Even if it is overestimated during the period of low discharge, the seasonal trend remains correctly addressed. However, as we plan focus the study on the lower limnic part the salinity-derived dispersion coefficients will not be very relevant for our reasoning. We will follow the reviewer's advice and remove this graph. Regarding the plots of the nitrogen-salinity relationship, please see our answer below to "Page 10 …". We will consider showing production rates for the focal area.

*Page 8 Line 232: where shown?*

The model performance regarding simulation of inorganic nitrogen species is decribed in detail from Page 8 Line 247 until Page 9 Line 269. There we refer to Figs. 4, 5, 8 and 9 illustrating the comparison model vs. observations.

*Page 9 Line 248-249: earlier you say that denitrification is included in the model.*

Yes, it is included in the model. The rates are mentioned in Sec. 5.1 (Page 14 Lines 451-456) with illustration given in Fig. 13d.

*Line 271 ff: Regarding the rather poor agreement of model salinity and temperature, the oxygen values should not be validated using saturation. Dissolved oxygen concentration should be used instead to avoid any offset due to incorrect salinity and water temperature.*

We would prefer to use the saturation, to see the over- or under-saturation and compare the saisonal data. However, we will provide a comparison model vs. observations in units mmol $O_2$ $m^{-3}$ for the supplementary material.

*Line 279: What means high accuracy? This comment relates to the entire validation section, both physical and biogeochemical: analysis of the comparison between observational data and model results seems to be done by plotting and "visual" judgement on the goodness of fit.*

We will provide further skill assessment of model performance both physical and biogeochemical in the revised version of the manuscript.

*Page 10 I don't see an additional benefit for model validation from comparing mixing curves and I there-fore recommend leaving out this section.*

This nitrogen-salinity relationships show very nicely the source-sink behavior of the treated herein nutrient. The issue of conservative or non-conservative mixing is central to the Elbe estuary literature which is why we would prefer to include respective plots when presenting a modified modelling approach. In addition these figures illustrate the good overall agreement between model and observations when looking at the estuary in isohaline coordinates. However, we might think of reducing the number of plots to one per nitrogen.

*Page 11 The entire chapter 5 has a structure which makes it hard to follow a central theme.*

We agree that our system description needs better structuring. We will separate results and discussion section in order to increase readability of the manuscript.

*Line 326: Announcing to explain the "full spacio-temporal dynamics of the estuarine ecosystem" is a very big promise and I strongly recommend being more moderate.*

We will change "full" for something else like "better spatially resolved" (in comparison with earlier works).

*Line 348-359: First, the model used in this study has not been validated for grazing or grazers. Second, the reference given (Schöl et al. 2014) is weak because the conclusion presented therein only bases on model results. Therefore, findings regarding grazers have to be validated, also to sustain the conclusions drawn from it later on. Observational data (counts of organisms, for ex-ample) might exist at the WGMN Hamburg or at BfG for the harbor area. Moreover, the model has also not been validated for phytoplankton. High-quality phytoplankton data are available, especially for the area that shows high model diatom concentration in Figure 10 (a).*

We agree that this validation is missing. However, observational evidence of grazers in the area of the port of Hamburg (main channel) has been published (Schoel et al. 2008, in German). The same publication shows that chlorophyll- a and grazer concentrations are anti-correlated in the along-channel direction, very similar to what we have presented as results of our modelling study. That is why we assume that grazing of phytoplankton in Elbe estuary is a valid approach. We will add the above publication to our reference list. We will also follow the reviewer's advice and provide validation of the model against the available chlorophyll-a measurements in the area of the Elbe river bifurcation.

*Page 12 Line365-367: The authors don't prove this statement.*

We reformulate this statement: Our results confirm a change of biogeochemical regime where the change in channel geometry occurs (i.e. we do not contradict previous works), however we demonstrate that changing depth is not the only possible reason but the horizontal geometry plays an important role with harbor basin opening a spatial niche for increased respiration, sedimentation, hypoxia and remineralisation.

*Line 367ff: The relation to the text in the first section of this paragraph isn't clear.*

We agree that this statement is misplaced here. We will either remove it or amend it to the conclusions.

*Line 376 and following two sections: When focusing to such a small region compared to the entire model domain, the authors should provide better validation for this specific area. A system-wide M2-comparison, which also shows its highest deviations in the focus region (roughly km 620 in figure 2(a)), is not sufficient. In addition: lacking validation for phytoplankton and grazers, see above.*

The dynamically important aspect is the gradient in kinetic energy between the tidal channels and the narrow and shallow basins. This gradient is an obvious consequence of the local geometry. Following the reviewer's advice we will provide better validation of the local water levels. This data is available.

*Page 13 Line 415 – Page 15 Line 476 This section can be shorted substantially because it contains many well-known and well-described processes and relations (contained in the references of the paper). The three – zone - description in the last section (Line 470 – 476) has first been described by Schroeder (1997). New aspects worked out by the authors are not clear.*

We will shorten this section and will make better use of references.

*Line 458: where is fig 12e?*

We change Fig. 12e as Fig. 12a, which shows the nitrate concentration at the station inside the harbor basin.

*Page 15 Line 478 – Page 16 Line 512 What is new &relevant in this section?*

This section describes the effect of inter-annual variability of the forcing onto the estuarine ecosystem taking into account the increased in comparison with earlier studies resolution of the complex channel geometry and associated increased spatial resolution of the coupled physical-biogeochemical processes. The ecosystem response to river discharge is very relevant both to scientific discussions as well as management and policy makers. We see both novelty and relevance given.

*Page 16 Chapter Conclusions Line 514 –522: Everything that is mentioned here has been shown already several times, for the Elbe and for other estuaries. A 3D model that is able to simulate all these processes is also not new but more or less common (see the comment on the abstract).*

In the revised manuscript we will better expose he benefits of the unstructured modelling and its relevance in the coupled physical-biogeochemical context. The revealed by our work hot spots of heterotrophic decay in the Elbe estuary can be a relevant aspect for estuaries worldwide, both in natural configurations or due to port construction. Using 1D, 2D or even structured curvilinear 3D models these processes could not be simulated in a generic way (i.e. without local parametrization of the physics and biogeochemistry) revealing the same systemic features. We will add corresponding statements to the revised manuscript.

*Line 523 – 528: Validation missing*

The validation is given for the main channel (Figs. 8,9 and S-1, S-2). In fact our simulation reproduces the nitrogen species distribution and magnitude with good accuracy which supports the predicted by the model organization of heterotrophic decay in the port area.

*Line 530 – 540: This section is more a description. What is the important conclusion out of it?*

This sections wraps up the results of the study. The important conclusion is that most of the time, i.e. given average or near-average river runoff, the system reveals a persistent compartmentation. However, the hindcast demonstrates that during the June 2013 flood event the usual and known compartmentation collapses being restored in the aftermath of the flood event.

*Line 542 – 543: This has already been shown for different estuarine systems worldwide.*

We will provide information on related studies in the revised manuscript. We will also better expose the relevance of using an unstructured 3D model and the importance of resolving narrow channels and complex channel geometry in estuarine ecosystem modelling.

*Line 543: From my point of view the importance of the harbor is the most interesting aspect that remains. However, I miss: 1. Better validation, 2. more focus, 3. analyzing hydrodynamic aspects as well (e.g. larger water volume in the harbor compared to section upstream), 4. comparison to other similar systems (like Guadalquivir-Sevilla, Loire, Humber or other estuaries containing large ports).*

We are grateful for this comment. We agree that resolving the harbor in this modelling study represents the key novelty. We will streamline the manuscript to better present this key aspect of the work. We will also provide better validation, covering the entire period of model integration and assess model performance using statistical measures. We will consider deepened analysis of the hydrodynamical aspects concentrating on the focal area of the study (channel bifurcation/harbor). We agree that…

*Figures There are many figures. Some figures (like 8, 9 or especially 12) are not easy to understand. Some panels (especially Figure 13) are very small. Check for Figure 2(b) to (e) if the description of x and y axis and the conclusion that in the model salinity intrudes too far into the estuary is consistent. There is no white dashed line visible in Figure 10.*

We will reduce complexity of figures 8,9 and 12, increase panel size were needed and add white dashed line to panels in Fig. 10. We agree that the number of figures hould be reduced. Therefore we might change part of the visual comparison between model and observations in Figs. 4,5,6,7 as tables giving ranges and basic statistics.

*Technical correction There is a plenty of typing errors, careless mistakes and wrong word order, some obviously due to copy& paste. I'm not a native English speaker, and therefore I'm not in the position to give suggestions for language corrections. However, I have the impression that language, including grammar and use of uniform tenses, needs substantial improvement.*

We will consider correction by a native speaker of the manuscript text prior to submitting the revised version of the manuscript in order to improve the language.

**Additional references:**

Hillebrand, G., Hardenbicker, P., Fischer, H., Otto, W., & Vollmer, S. (2018). Dynamics of total suspended matter and phytoplankton loads in the river Elbe. *Journal of soils and sediments*, *18*(10), 3104-3113.

Schöl, A., Blohm, W., Becker, A. , Fischer, H. (2008). Untersuchungen zum Rückgang hoher Algenbiomassen im limnischen Abschnitt der Tideelbe. https://www.bafg.de/DE/08_Ref/U2/01_mikrobiologie/algen_tideelbe.pdf?__blob=publicationFile

Weilbeer, H. (2014). Sediment transport and sediment management in the Elbe estuary. *Die Küste, 81 Modelling*, (81), 409-426.

---

## Author Comment (AC2) · 23 Sep 2019

**Answers to Referee 2**

*There are many major issues to this submission:*

We thank the referee for his/her comments. In the following we respond to the individual points. Reviewer's comments are written in italics, while authors' answers are in kept in plain font.

*1) There is no/little hydrodynamic calibration. Authors need consider to submit two manuscripts: one for hydrodynamic and one for water quality dynamics.*

The Elbe set-up has been derived from a larger set-up of the German Bight (Stanev et al. 2019). The model area of the recent set-up is completely contained in the already published set-up, they share the same topography and – most important – the Elbe set-up is hydro-dynamically driven by the German Bight set-up. A separate publication on the physical estuarine dynamics would be necessary and justified in case the processes involved or parametrization have changed, which is not the case. However, the model integration period has been extended from several months (01.01.2013 to 31.08.2012) to two years, 01.01.2012 to 31.12.2013. Therefore, more observational data has become available for the time of model integration. Also cruise data including both physical and biogeochemical data have been made available to us. In the submitted paper we have made use of the newly available data (Fig. 2 b-e). To partially follow the reviewer's advice we will provide additional validation using stationary salinity and temperature measurements which we will present either in form of diagram or tables with ranges and basic statistics. In this way we will increase credibility of the physical simulation in the focal area.

*2) Why authors didn't calibrate the water quality for the bottom part, particularly to the oxygen? Ammonia simulation is a little different from the observed one, any justification?*

We find the model performance regarding water quality, for example oxygen saturation, is very convincing. In particular the good agreement of simulated oxygen saturation with the observed values particularly at Hamburg station during summertime (Fig. 6c) demonstrates that the predicted by the model increased bottom respiration in this area contributes to the realism of the simulation.

Regarding the ammonia simulation it would be good to know exactly to which "little difference" the reviewer refers to. Fig. 5c, d, f show a mismatch between model and observations. The model underestimating ammonia levels is likely to be due to underestimated water temperature. Therefore the most likely explanation is non-optimal boundary forcing. The ammonia variability is however very similar in model and observations (Fig. 5c, d). To better illustrate and specify the agreement between model and observations we will provide ranges and basic statistics for the stations measuring biogeochemical variables in Figs. 4-7.

*3) The model set up and data description is very weak, and need a lot of work to this part. Again, authors need consider to split this manuscript into two manuscripts. Why choose year 2012 and 2013?*

We have written in model description (pages 4-5, lines 99-108 and lines 122-141), that the model framework combines two established models, SCHISM (Zhang et al. 2016, Stanev et al. 2019) and ECOSMO2 (Daewel and Schrum, 2013). These models have been described in details in the previous works. However, there is a novel aspect to this particular set-up which is the coupling through the FABM (Bruggeman and Bolding, 2014). The coupling software also accounts for the simulation of the interaction between the water column and the organic

sedimentary layer. To partially follow the reviewer's advice, we will add a decent description of the coupling through the FABM. We will also provide a table with the parametrisations used for the biogeochemical model.

To be consistent with the earlier works including the Elbe estuary (Stanev et al., 2019), we start the simulation in the same year (2012). We integrate it for another year (2013) in order to establish a data-set embracing the seasonal to inter-annual variability of the biogeochemical processes. Furthermore we have observations available for these two years which allowed us to perform the necessary biogeochemical validation. Therefore we find the chosen period well-justified.

*3) This study is very local, and there is no linkage to broad area? What is the contribution of this work the research community? The questions is pretty local, and not novel? Authors even didn't fully answer the questions of introduction part.*

We agree with the reviewer that linkage with other estuarine studies needs to be improved. Deepened comparison with other biogeochemical studies on estuarine ecosystem and the relevance of our study for the global situation of estuaries will be provided in the revised manuscript.

The novelty of our study is that the unstructured mesh has been used to resolve the 3D coupled physical-biogeochemical processes in the narrow, curved channels and small basins of the Hamburg port area. Our simulation reveals the hot spots of sedimentation, hypoxia and remineralisation. These are located in particular in the side basins and channels of the harbor area which is why our study reveals a novelty compared to previous modelling studies. In the revised manuscript we will better clarify the novel aspects of our study. We will also provide its relevance for similar estuarine configurations worldwide.

We agree that the answers to research questions need to be more complete. We find this comment linked to the next one "*however authors want to cover everything".* Therefore, we propose to better streamline the manuscript focusing on the impact of biogeochemical processes in the port area onto the estuarine nitrogen cycling.

*4) The mixing diagram was used by Jiang and Xia, 2018 and isn't new. This study is mainly for nitrogen dynamics, however authors want to cover everything. It is a little bit difficult to follow, and authors need think how to make a nice flowchart to this manuscript. Overall, it reads like a modeling or technical report. There are many minor issues, however I would like authors to take care of major issues now.*

We use the mixing diagram to a) validate the along-channel distribution of the inorganic nitrogen species over the estuarine salinity gradient (in isohaline coordinates), b) characterize the mixing behavior of these species along the Elbe River. The method is of course not new (see our references), however there is no such extensive demonstration of mixing behavior of nitrogen species for the Elbe system. As the paper focusses on the nitrogen cycling we consider its mixing behavior a central aspect.

The referee is right that we need better streamlining of the manuscript. As written in the answer to the previous comment, we plan to focus the study onto heterotrophic decay confined to the harbor basins and side channels and its impact onto the estuarine nitrogen cycling. This will lead to a more balanced manuscript, where new scientific inside arising from the more complete representation of processes in the port area will be set in relation to better known

spatio-temporal organisation of nutrient cycling in the main channel downstream from the port area.

*Jiang, L. & Xia, M. (2018), Modeling investigation of the nutrient and phytoplankton variability in the Chesapeake Bay outflow plume. Progress in Oceanography, 162, 290-302. doi: https://doi.org/10.1016/j.pocean.2018.03.004*

We thank the referee for recommending this useful reference.

---

## Author Comment (AC3) · 15 Oct 2019

**Answers to Referee 1**

We thank Ingrid Holzwarth for her dedicated effort to read and comment on our paper. In the following we will give our point-by-point answers to the individual comments or paragraphs of her review. Reviewer's comments are written in italics, while authors' answers are in kept in plain font. We attach additional or changed for the revised version of the paper figures and tables at the end of this document.

General comments My overall impression is that the manuscript is a draft with a collection of several interesting topics (importance of a 3D model to capture estuarine biogeochemistry, nitrogen cycling, hydro-dynamic transport times, grazing, influence of river discharge, small-scale hydrodynamic pat-terns influencing the system-wide biogeochemistry) but no consistent story line and no clearly expressed conclusion in the end. My general advice is to focus on one aspect from the collection of interesting aspects above and present it in a clear and concise way.

Authors: We are glad that the reviewer discovers several interesting aspects to our work. We agree that the paper needs further streamlining and focusing on the most relevant aspects. At the same time we would like to remind here that a complex system such as the coupled physics and biogeochemistry of the Elbe estuary require an integrate description. An excess of fractioning the integrate system into sub-systems and aspects increases the danger of overlooking important interconnections. For example we could not explain a spatial shift in a nutrient pattern without accounting for advection/diffusion of salinity and freshwater discharge, nor could we do so without accounting for organic inputs at the weir or diatom growth in the shallow limnic part. Our work demonstrates that the specific spatio-temporal organization of the nitrogen turnover in Elbe estuary depends on the physical and biogeochemical processes around the bifurcation of the river near Hamburg and specifically in the shielded from the tidal currents harbor basins and side channels. Following the reviewer's advice, in the revised manuscript we will focus on how the specific dynamics in the area of the Elbe inland delta region including the Hamburg port affect the nitrogen cycling in Elbe estuary.

**The manuscript presents a work very specifically done for the Elbe Estuary without discussing the global situation or the situation in other similar, or different, estuarine systems. Almost no literature apart from specific Elbe Estuary literature is included**

Authors: We agree that the linking between our present work and its relevance for other estuarine systems needs to be improved. We will provide a better representation of studies on other estuaries or coastal systems as well as a statement on the global relevance of our study in the revised version of the manuscript. We will add a paragraph to the discussion section saying:

"Elbe estuary shares the fate of many coastal and semi-enclosed systems worldwide where oxygen depletion and hypoxia arise predominantly due to a combination of two factors: high organic inputs and vertical stratification (Kim et al., 2010; Zhou et al., 2017) or high organic inputs and high sediment concentration (Ruiz et al. 2013; Zhou et al. 2017; Lajaunie-Salla et al., 2017). Also, like other similar systems Elbe estuary reveals increased ammonia levels associated with the oxygen minimum zone (Kim et al., 2010; Holzwarth and Wirtz, 2018; Zhang et al. 2018). In our simulation of Elbe estuary, both sedimentation of organic material and isolation of eutrophic water lead to hypoxia, release of ammonia and denitrification. Our work reveals that highest levels of hypoxia and ammonia concentration emerge in the small side channels and harbor basins of the Elbe inland delta. Good agreement between observed and simulated longitudinal pattern of ammonia suggests that the shielded from strong tidal currents

small side channels and basins indeed affect the nitrogen cycling of the main channel. We thus show that small features deserve to be resolved in modelling and need to be taken into account by monitoring. We also show that lateral segmentation may have effects similar to vertical stratification."

**Though I agree on the necessity to use a 3D model for analyzing estuarine biogeochemistry, this point is not worked out well: (1) The argumentation for the necessity of a 3D model fails.**

Authors: A 3D model is necessary in case axial gradients, lateral exchange processes and stratification of physical and biogeochemical properties interact and control one another. The coupled physics and biology in the main channel of Elbe river could and have been analyzed in a simplified manner using 1D or 2D representations of the river geometry. In the area of the Elbe inland delta however, stratification, lateral gradients and exchange feed back onto the longitudinal dynamics (as seen for example in both observations and model from the estuarine ammonia maximum, a sharp peak located about Elbe-km 625, Fig. S-2). That is why a 3D model is necessary to increase the realism of the simulation beyond what has been achieved in the past modelling. This knowledge might also be useful for estuarine management. We find that the benefits put forth by the unstructured 3D coupled physical-biogeochemical modelling in this particular case are of general scientific interest and deserve to be made public.

**(2) The authors make little use of the additional information they get from their 3D model compared to information from measurements or a 1D/2D model.**

Authors: While aspects of the advantage of the highly resolved model have been partially addressed by our analysis (e.g. along-section profiles in Fig. 11) we agree that additional information retrieved specifically form a unstructured 3D model will improve our manuscript. Following the referee's advice we will add time-averaged maps of the state variables most relevant to the heterotrophic decay around the channel bifurcation and port area to the manuscript (please see end of document, Fig. D-1).

Apart from the issues above, the model (physical and biogeochemical) is not sufficiently validated for the argumentation later on. While I see nitrogen and nitrogen species well-validated, all other quantities are not. In any case, the physical model need better validation, and – given the poor agreement shown for salinity – probably some more effort in setting up a better physical model is required.

Authors: The Elbe set-up is a derivative of an already published set-up (Stanev et al. 2019) and it is also hydrodynamically coupled to the latter (see page 4 line 114-116). Therefore, the hydrodynamical model of the herein presented coupled physical-biogeochemical model has actually been validated for the Elbe region and underwent the peer review process. However, the period of integration has been extended and the simulation has been compared to new observations available, i.e. the ship-borne measurements. We admit that simulation of salinity for certain periods in 2012 does not match expectations. There can be several reasons such as for example non-optimal boundary forcing or overestimated horizontal diffusivity. Both, operational model development and control of physical diffusion are developing topics of the modelling community (Graham et al, 2018; Ralston et al. 2017). We agree that additional validation for salinity, temperature and entangled with the nitrogen turnover biogeochemical properties using the available stationary observations.

Please find at the end of this document the following tables:

Table 1: Comparison of observed and simulated tidal amplitudes of water elevation for five major tidal constituents for a number of gauges along Elbe estuary using the online available data (kuestendaten.de) for the period of model integration.

Table 2: Comparison of observed and simulated surface salinity and temperature at a number of stations along the Elbe estuary using the online available data (www.kuestendaten.de).

Table 3: Comparison of observed and simulated surface nutrient concentrations at a number of stations along the Elbe estuary using the FGG Elbe data also used for forcing the model at the land boundary and presented in the first submission in Figs. 4, 5, 6, 7.

Table 4: Comparison of observed chlorophyll and derived from simulated primary producers chlorophyll at two stations at the upstream end of the bifurcation of the Elbe river and station Hamburg (km 629) using data from Hamburg authorities (www.gateway.hamburg.de) and from FGG Elbe (www.fgg-elbe.de).

Specific comments Title It is not clear what the authors mean with "joint perspective". The manuscript presents a model-ling study with the use of observational data for model steering and validation, which is the usual way to do.

Authors: We agree that in particular the second part of the paper is essentially a modelling study. On the other hand we use observations for complementary analysis of nitrogen turnover processes (Figs. 8, 9). As a compromise we propose to change the manuscript title as "Nitrogen cycling in the Elbe estuary: Observations and numerical modelling".

**Abstract Line 7-8: observations are only used for steering & validation here.**

Authors: It is correct that we use observations for boundary forcing and validation, however complementary analysis of estuarine nitrogen cycling is based on observations. For example on page 10 Line 307-309 we have written "In autumn, observations revealed concave bending at low salinities, indicating a nitrate sink in this area (Fig. 8c) or gradually increasing nitrate concentrations in the river Elbe (Fig. 4a)."

Line 9: "a coupled physical-biogeochemical 3D unstructured model, applied for the first time to the estuarine environment". It is not clear, what the authors mean: if the sentence relates to the specific model they use (Schism – ECOSMO) it might be true but is nothing new and remarkable in the abstract. However, the expression could also be misleading: I'm not sure if the authors mean that it was the first time ever that a coupled model has been applied to an estuary. In this case, the authors are referred to Cerco et al. (1993), Wool et al. (2003), Wild-Allen & Andrewartha (2016), Lajaunie-Salla et al. (2017) – to name only some.

Authors: We admit that our formulation is not optimal. Our statement intended to stress two aspects: 1) we present the first unstructured coupled 3D modelling using realistic bathymetry of Elbe estuary, 2) for a first time the ECOSMO biogeochemical model is applied in the estuarine environment. In the revised abstract we will simply state that we use 3D unstructured coupled physics-biogeochemistry model and realistic geometry.

We thank the reviewer for recommending above references. The cited studies are certainly relevant for our study as they employ 3D coupled physical-biogeochemical model in the

estuarine context. In the context of discussing the novelty of our modelling effort, we would like to point out that non of these modelling studies used unstructured mesh to resolve small scale features and curved narrow channels. However we find useful information in particularly in the more recently published of the above cites studies and we will integrate their findings into our reasoning.

**Line 13 – 21: - Apart from the validation for nitrogen, model validation for all other physical and biogeo-chemical quantities is not sufficient to support these findings. - What is the new& important finding and what are its consequences?**

Authors: The new and important aspect of our work resides in sharpening the understanding of nitrogen cycling in Elbe estuary by use of an unstructured mesh in the context of 3D coupled physical-biological modelling including an organic sediment tool. This tool is very capable to address the biogeochemical functioning of the Elbe estuary because it enables the integration of coupled dynamics in narrow, curved channels and harbor basins of Elbe estuary. Our modelling study reveals the hot spots of heterotrophic decay in the area of the Elbe inland delta. To our knowledge, the past modelling studies have not elaborated deeply on this aspect of the Elbe estuary ecosystem dynamics, and they have not been able to reproduce crucial patterns like oxygen minimum and ammonia maximum with the same accuracy. To the best of our knowledge also observational studies on Elbe estuary have not systematically probed distributions of nutrients, oxygen and chlorophyll in at the same time in small channels, harbor basins and main channel during a period of several years and with a spatial and temporal resolution comparable to what we present in this study. Our work smoothly connects to important previous studies like those of Schroeder (1997), Holzwarth and Wirtz (2018), and Sanders et al. (2018), adding important details to the community's understanding of the spatiotemporal distribution of Elbe ecosystem biogeochemistry. Following the reviewer's advice, we will better clarify the novelty of our work and its consequences for the understanding of the ecosystem in the revised manuscript. Please find additional validation of physical and biogeochemical properties in Tables D-1, D-2, D-3 and D-3 below.

**Page 2 Line 31: The reference Jickells et al. 2014 is not appropriate for the statement.**

Authors: We will look for an appropriate reference to or change the statement. A possible reference would be Weilbeer (2014), who mentions ongoing maintenance dredging which does however not imply morphological changes in the future.

Line 38-39: If oxygen depletion is mentioned later on, it should be mentioned that depletion was most severe during this period.

Authors: We will add this information to the revised manuscript.

Line 61-62: Sentence unclear: in x direction (main flow direction), all of the above mentioned models describe the gradient between the weir and the German Bight.

Authors: We reformulate "These studies however did not resolve the entire transition between the tidal weir and the oceanic background while assuming vertical (and some even lateral) homogeneity of the gradient-rich estuarine environment."

"With the onset of coupled unstructured modelling it is possible to go one step further resolving the complex branched geometry, vertical and lateral gradients and exchange processes, all of which we may expect to feedback onto the longitudinal physical-biogeochemical dynamics between the tidal weir and the ocean background."

Entire last paragraph: the models mentioned before all focus on oxygen, the authors focus on nitrogen (or meso-zooplankton?). The authors should consider that this might be a completely different story.

Authors: We are grateful for this comment. It is correct that the cited studies focus on oxygen. In our model, nitrogen and oxygen concentrations are dynamically linked through production and degradation processes. We will clarify the interdependency of nitrogen and oxygen in the revised manuscript. We will also add a statement about the specific approach implemented in ECOSMO2 in comparison with the previous studies on the Elbe system.

We reformulate "The oxygen and nutrient dynamics in the tidal river have recently been successfully addressed by several models."

as

"The oxygen dynamics in the tidal river have recently been successfully addressed by several modelling studies."

We reformulate "The above studies showed that even with spatially and biogeochemically simplified models it is possible to coherently simulate major estuarine processes and answer question regarding climate impacts on estuaries or estuarine management. These studies however did not resolve the entire transition between the tidal weir and the oceanic background while assuming vertical (and some even lateral) homogeneity of the gradient-rich estuarine environment. We hypothesize that such simplification is not justified."

**as**

"The above studies showed that even with spatially and biogeochemically simplified models it is possible to coherently simulate the major spatial and temporal characteristics of Elbe estuary oxygen dynamics and answer questions regarding climate impacts on estuaries or estuarine management. These studies however did not elaborate on the details of the cycling of a major nutrient, nitrogen, taking into account realistic geometry, lateral and vertical exchange like sinking, burial and resuspension of the nutrients. Our work shows that it is worthwhile to increase the realism of the simulation taking into account 3D coupled physical-biogeochemical processes both in the main channel as well as in the branched geometry of the Elbe inland delta. "

Page 3 Line 70-74: limiting to one or two questions would probably help to give a more concise and focused article.

Authors: We agree that the paper needs substantial streamlining.

Line 75-81: the stations of WGMN Hamburg (Blankenese, Seemannshöft, Bunthaus) should also be taken into account. They provide high-quality data in an area where later on important conclusions are drawn from model results. And they give a differentiation of phytoplankton between diatoms, green algae and blue-green algae. Authors: Some of this data are indeed available for the given period. Blankenese data is currently not available according to the online tool. The chlorophyll-a data raised by Hamburg authorities (WGMN) available for stations Bunthaus (km 609) and Seemannshöft (km 629) have been used for validation of variability of the primary producer concentrations, please see Table 4 at the end of this document. Table 4 shows that the mode predicts the decrease of average primary producer concentrations between the upstream and downstream end of the main channel bifurcation (i.e. between km 609 and km 629). R2 and Willmott score computed for the three stations reveal that model performance is comparable to what other authors have presented for chlorophyll (e.g. Wild-Allen and Andrewartha, 2016).

Line 80: Use boundary "values". At this point I miss the information about the location of the model boundaries.

Authors: The landward model boundary is at Geesthacht (Fig. 1). The stationary measurements at the tidal weir at Geesthacht (Elbe-km 585.9) were used to force the model at its landward boundary. We have specified this on page 5 line 154-156 of the manuscript. We will remove the statement about model boundaries at line 80 to avoid confusion at this point of the manuscript.

Line 85: it should be mentioned that measurements are taken approx. 1m below the water surface. No vertical (and no lateral) gradients available.

Authors: We will add a statement "Observations have been sampled approximately 1 m below the water surface with also lateral resolution not available."

**Line 92-95: references missing**

Authors: The numbers have been compiled from the data set used for model forcing at the open boundary. We will add this statement to the revised version of the paper.

Page 4 Line 96-97: unclear: were climatologic averages used? Daily values? Is the same method used for all nutrient species?

Authors: We used daily values of river runoff (cf. line 91). Nutrient observations were sparser in time. We generated a daily time series of observed nutrients using cubic spline interpolation (please see time series used for forcing below, Fig. D-2). We used the same method for all nutrient species.

**Line108: What is GCOAST? Is it important information?**

Authors: The "Geesthacht COAstal model SysTem" (GCOAST) refers to a high-resolution modelling tool that couples ocean, wave and atmosphere dynamics. It is currently under development at Helmholtz-Zentrum Geesthacht. This could be relevant information because the modelling of estuarine ecosystem dynamics presented in this study is part of a greater effort to develop a modelling tool for research into the entire land-ocean transition zone.

Page 5 Line 128: What exactly are the state variables? What is their functional relation? The authors say that organic matter decomposition was very important (which is true), so was the ECOSMO2 setting changed from the references given before? If it is not changed, this information should be clearly given here. In case it is changed (also in case certain parameters are changed), the model functions and their parameter values have to be described in more detail.

Authors: The state variables are nitrate, ammonium, phosphate, silicate, oxygen, diatoms, flagellates, cyanobacteria, micro-zooplankton, meso-zooplankton, dissolved organic matter, biogenic opal, detritus, sediment, sediment phosphate, sediment silicate. A detailed description of the biogeochemical model we use, ECOSMO2, is given in Daewel and Schrum, 2013. The state variables and processes remained unchanged. The parametrization remained largely unchanged. A table with state variables and parameters used will be added to the supplementary material of the revised paper. The interaction between the water column and the sedimentary layer remained unchanged as well, it has however been implemented into the FABM framework. We will add a description of the FABM module to the revised manuscript.

We reformulate page 5 line 128

"Therefore model formulation for organic matter decomposition is of utmost importance."

As

Therefore model formulation for organic matter decomposition is of utmost importance. The herein used biogeochemical model ECOSMO2 (Daewel and Schrum, 2013), predicts heterotrophic turnover of organic matter allowing for decay and remineralisation both in the water column as well as in a sedimentary layer"

**Line129: Assumes Redfield ratio in phytoplankton, I guess?**

Authors: Yes, this is correct. Redfield ration is used throughout the model, please sea line 130 of the manuscript.

**Line 133: What kind of feedback is meant here?**

Authors: We will remove this phrase from the manuscript.

Line 134: Unclear: first, the authors say that the sediment is a mere storage. Then they say, mineralization and nitrogen release was possible.

Authors: Yes, it is right that the sediment is a storage, i.e. a reservoir of nutrients at the bottom. This reservoir gets filled by sedimentation. It can be depleted by remineralisation and resuspension of the nutrient. Probably the word "mere" was misleading here and we will remove it.

Line 140-141: Holzwarth and Wirtz 2018 say: turbidity is so high that depth plays a secondary role. What is the value for the background attenuation coefficient used here? In case it is the same value used for the North Sea it will certainly be too low. See comment above: biogeochemical model description is missing.

Authors: The background attenuation coefficient was 0.05 m-1. We will add a table with the coefficients used to the revised manuscript.

We reformulate line 140-141 the phrase

"In effect light attenuation is controlled predominantly by water depth, which has been shown a valid approach for the area of Elbe estuary in the study of Holzwarth and Wirtz (2018)."

as

"In effect the longitudinal change in light climate is predominantly controlled by turbidity and channel depth (Schroeder, 1997; Holzwarth and Wirtz, 2018), where herein the turbidity-induced light attenuation is accounted for by self-shading of plankton. "

**Line 156: Which station exactly? What is the temporal resolution of the observational data?**

Authors: We refer above line 91-92 to daily discharge measured at Neu Darchau. This will be clarified in the revised manuscript.

Line 159: Is the conversion factor in line with the conversion used in other Elbe studies (some of them were already cited)? If not, why different values were used? Entire paragraph with Geesthacht boundary values: please provide a plot containing the time series of the used boundary values.

Authors: Hillebrand et al. (2018) give estimates of the Chl(a)-to-carbon ratio in Elbe river ranging from 18.2  $\mu$ g Chl-a/mg C during winter months up to 32,3  $\mu$ g Chl-a/mg C during summer months. Herein cited studies used values between ~27  $\mu$ g Chl-a/mg C and ~ 50  $\mu$ g Chl-a/mg C (Schöl et al., 2014; Holzwarth and Wirtz, 2018). Given that in our model C:N ratio equals 106:16 we effectively used a conversion value of ~ 20  $\mu$ g Chl-a/mg C. We used this value following the applied by Zhou et al. (2017) 1,58 g Chl-a/mol N. Assuming Redfield ratio of C:N this value appears to be in the range of Chl-a/mg C proposed by Hillebrand et al. (2018) for Elbe river.

The forcing time series used at the tidal weir are presented in Fig. D-2 at the end of this document. We will move it to the supplementary material of the revised paper.

**Page 6 Line 164: Please clarify: 2012 was used as spin-up period?**

Yes, 2012 was ran as spin-up and then repeated as actual herein presented simulation. We will clarify this in the revised manuscript.

Line 171: Why can the authors assume, based on M2 tide, that water levels deviate less than 10 %? Besides: - I consider 10 % as far too poor, given a water depth more than 15 m in large parts of the estuary. This would imply a substantial error in water volume. - More information on tidal analysis for observational and modeled data (mean high and low water, tidal range, slack water times) would be helpful to assess the quality of the physical model, essentially including an analysis of current velocities.

We agree that based on assessment of model performance regarding the M2-tide only it is unlikely to assure that errors are less than 10%. Model performance is actually better (see also Stanev et al., 2019). For the revised article we have prepared comparison between observed

and simulated five major tidal constituents amplitudes of water elevation (please see Table 1 below) giving a more accurate presentation of model performance in respect to tides.

**Line172: "ventilation" might be misleading in this context (could be understood as "reaeration")**

By "ventilation" we mean "forcing and stirring of the estuary by tides". We agree it could be misunderstand in terms of reaeration. We will reformulate this phrase accordingly.

Line 174-175: The stationary monitoring data of the water and shipping administration, which are distributed along the salinity gradient, from Hamburg to Cuxhaven should be used for validation. Using these data, a time series comparison for the entire simulation period, including basic statistics, is possible. This would give a much better impression on the quality of model results than comparing 4 single transects. Same for temperature.

We agree that stationary measurements of salinity and temperature should be taken into account. We will provide corresponding plots and basic statistics in the revised version of the manuscript (please see Table 2 towards end of document).

Line 176: The deviation in salinity is quite high. Given the really high overestimation in the October 2012 profile (up to 15 psu, if I understand it correctly) I strongly recommend to include the stationary measurements, see above. Either the ship-based observational data are bad, or the physical model needs improvement.

Authors: Following the referee's recommendation we have used stationary observations to validate model performance regarding salinity (Table 2). According to this assessment bias between model and observations in the center of the salinity front is up to ~7 psu. By km 650 observations and model results both show ~0.5 psu on average (Tab. 2). Overestimation of salinity intrusion mainly happens at below-average discharge, whereas at higher discharges model attains good realism (for example at Cuxhaven (km 725) during the first half of year 2013, Fig. D-3 below). R2 and Willmott score (WS) are 0.57 and 0.39 on average, respectively (taking into account the four stations in Table 2), which is comparable to what other authors of coupled estuarine modelling have presented (for example in Derwent estuary Wild-Allen and And rewartha (2016) attained  $R^2 = 0.31$  and WS = 0.52 for simulated salinity). Modelling of the salinity dynamics and salinity intrusion is subject to ongoing efforts and discussions in the estuarine modelling community (Fringer et al., 2019). Wrapping up discussions at an international workshop on coastal modelling, these authors identify challenges such as good forcing data and control of numerical diffusion in presence of sharp fronts or bathymetry varying at grid-scale (both of which is the case in Elbe estuary). Ralston et al. (2017) point out that implementation of higher-order advection schemes is a particular challenge to unstructured modelling which comes thus as a trade-off when such a model is used in order to resolve complex geometry such as narrow curved channels. In our study we focus on the limnic reach of the estuary where nutrient loads and turn-over are dominated by river discharge and marine influence is small. Towards the mouth of estuary both marine loads become more important and model performance improves (Table 2).

Line 181: I assume the authors refer to model results (please clarify, also in the figure text)? Given the poor agreement between observational data and model results I recommend not to show this figure until better agreement is proven.

Authors: Yes, here we refer to seasonal variation of simulated salinity. We will change Fig. 3a for a stationary time series showing seasonal variability of salinity in both model and observations at Cuxhaven (km 725, see below Fig. D-3). The study focuses on nitrogen cycling happening primarily in the limnic part dominated by riverine inputs. Thus errors in the position of the salinity front to not negatively impact on biogeochemical model performance and overall dynamics in this area.

Page 7 Entire paragraph on dispersion and tracer: I recommend proving good performance of salt transport (see above) before calculating a dispersion coefficient based on salinity. More: the relevance of the entire section is not clear to me. I recommend leaving it out completely and better spending more effort in the validation of the physical model. Besides, mixing plots for model data are unnecessary from my point of view: having a model, you can extract from your model results the production rates for all your state variables in your entire model domain. Then you clearly see (in numbers!) whether you have a biogeochemical activity or not. You can even see where it comes from. Maybe the authors can make better use of this information which is so far not used in their study.

Authors: The model-derived dispersion coefficient illustrates the seasonal modulation of alongchannel dispersion. Even if it is over-estimated during the period of low discharge, the seasonal trend remains correctly addressed. However, as focus of the study on the lower limnic part the salinity-derived dispersion coefficients will not be relevant to our reasoning. We will follow the reviewer's advice and remove this graph.

We still find the property-salinity relationships useful to give an overview of model performance and mixing behavior for the state variables central of to our study. They show clearly the great difference of concentration levels between the limnic part and the saline region. To partially follow the referee's advice we will reduce the usage of these plots, showing simple comparisons model vs. observations for nitrate, ammonium and oxygen saturation. Figures 8 and 9 of the submitted paper will be replaced by Figure D-4 shown at the end of the document.

**Page 8 Line 232: where shown?**

Authors: The model performance regarding simulation of inorganic nitrogen species is described in detail from page 8 line 247 until page 9 line 269. There we refer to Figs. 4, 5, 8 and 9 illustrating the comparison model vs. observations. In the revised manuscript Table 3 below will give basic statistical measures of model performance regarding inorganic nitrogen species.

**Page 9 Line 248-249: earlier you say that denitrification is included in the model.**

Authors: Yes, it is included in the model. The rates are mentioned in sec. 5.1 (page 14 lines 451-456) with illustration given in Fig. 13d.

Line 271 ff: Regarding the rather poor agreement of model salinity and temperature, the oxygen values should not be validated using saturation. Dissolved oxygen concentration should be used instead to avoid any offset due to incorrect salinity and water temperature.

Authors: We skill assessment of model vs. observations of oxygen concentrations [mmol m-3] in Table 3 towards the end of this document. In order not to repeat information given in the

tables, we will change Figures 4, 5, 6, 7 for a single figure showing time series of nutrient observations and chlorophyll as well as respective model results for station Hamburg (km 629), only. This new figure is given at the end of the document (Fig. D-5 herein).

Line 279: What means high accuracy? This comment relates to the entire validation section, both physical and biogeochemical: analysis of the comparison between observational data and model results seems to be done by plotting and "visual" judgement on the goodness of fit.

Authors: Please see tables 1-4 at the end of this document which provide skill assessment for physical and biogeochemical variables along Elbe estuary.

**Page 10 I don't see an additional benefit for model validation from comparing mixing curves and I there-fore recommend leaving out this section.**

Authors: This nitrogen-salinity relationships show very nicely the source-sink behavior of the treated herein nutrient. The issue of conservative or non-conservative mixing is central to the Elbe estuary literature which is why we have included respective plots when presenting a modified modelling approach. In addition these figures illustrate the good overall agreement between model and observations when looking at the estuary in isohaline coordinates. However, following partially the reviewer's advice, we have reduced the use of property-salinity relationships, showing diagrams covering the summer months during the period of model integration. Nitrate-salinity, ammonium-salinity and oxygen saturation-salinity relationships compiled from model and observations will replace Fig. 8 and 9, please see Fig. D-4 below.

**Page 11 The entire chapter 5 has a structure which makes it hard to follow a central theme.**

Authors: We agree that our system description needs better structuring. We will separate results and discussion section in order to increase readability of the manuscript.

Line 326: Announcing to explain the "full spacio-temporal dynamics of the estuarine ecosystem" is a very big promise and I strongly recommend being more moderate.

Authors: Our statement refers to limited in space and time representation of biogeochemical dynamics given by stationary data or transects presented in this section. In order to avoid misunderstanding we will change "full" for "better resolved in space and time than single transects or stationary data".

Line 348-359: First, the model used in this study has not been validated for grazing or grazers. Second, the reference given (Schöl et al. 2014) is weak because the conclusion presented therein only bases on model results. Therefore, findings regarding grazers have to be validated, also to sustain the conclusions drawn from it later on. Observational data (counts of organisms, for ex-ample) might exist at the WGMN Hamburg or at BfG for the harbor area. Moreover, the model has also not been validated for phytoplankton. High-quality phytoplankton data are available, especially for the area that shows high model diatom concentration in Figure 10 (a).

We agree that this validation is missing. However, observational evidence of grazers in the area of the port of Hamburg (main channel) has been published (Schöl et al. 2008, in German).

Schroeder (1997) presents grazing in Elbe estuary as a matter of fact, too. Further observational studies from coastal and estuarine environment prove that zooplankton does live in turbid environment given that phytoplankton, as identified by Chl-a measurements, is available (for example in Guadalquivir, Scheldt, and Ob estuaries: Ruiz et al. 2013; Muylaert et al., 2001; Arashkevich et al., 2010, respectively). Schöl et al. 2008 show that chlorophyll- a and grazer concentrations are anti-correlated in the along-channel direction, very similar to what we have presented as results of our modelling study. That is why we assume that grazing of phytoplankton in Elbe estuary is a valid approach. We will add the above publication to our reference list. We will also follow the reviewer's advice and provide validation of the model against the available chlorophyll-a measurements in the area of the Elbe river bifurcation (please see Table 4).

**Page 12 Line365-367: The authors don't prove this statement.**

**Authors: We reformulate this statement**

"We found however that the regime shift might not only me attributed to changed channel depth or vertical dimension only, and that a more complete understanding requires consideration of three-dimensional variability in time."

as

"Our results confirm a change of biogeochemical regime where the change in channel geometry occurs (i.e. we do not contradict previous works), however we demonstrate that longitudinal patterns do not only change depending on depth but horizontal geometry plays an important role with harbor basin opening a spatial niche for increased respiration, sedimentation, hypoxia and remineralisation."

We will use Fig. 11 in combination with a new figure, given below as Fig. D-1, to demonstrate that predicted by the model hotspots of heterotrophic decay and remineralisation are located outside the main channel and outside the deepest part of the local geometry.

**Line 367ff: The relation to the text in the first section of this paragraph isn't clear.**

Authors: We agree that this statement is misplaced here. We will either remove it or amend it to the conclusions.

Line 376 and following two sections: When focusing to such a small region compared to the entire model domain, the authors should provide better validation for this specific area. A system-wide M2-comparison, which also shows its highest deviations in the focus region (roughly km 620 in figure 2(a)), is not sufficient. In addition: lacking validation for phytoplankton and grazers, see above.

Authors: Fig. 2a give M-2 amplitude in the model along the channel axis whereas gauges are located at the shore. This figure is a compromise to give a comprehensive but not very exact overview of agreement between model and observations. We consider to remove observations from this figure and give detailed information on model performance regarding tides in a table, please see Table 1 below. The tidal amplitudes of five major tidal constituents measured at several gauges along the estuary and in the focal region have been compared to model results. Table 1 shows acceptable agreement between predicted and observed tidal amplitudes, in particular in the focal area (around km 629).

Page 13 Line 415 – Page 15 Line 476 This section can be shorted substantially because it contains many well-known and well-described processes and relations (contained in the references of the paper). The three – zone - description in the last section (Line 470 – 476) has first been described by Schroeder (1997). New aspects worked out by the authors are not clear.

Authors: Schroeder (1997) focusses on oxygen although his model includes carbon and nitrogen. His general findings are valid until today, however important details are missing. For example his simulation did not reproduce the ammonia peak at the lower end of the Elbe inland delta (station Hamburg in our manuscript, km 629). Also, Schroeder (1997) does not explicitly mention three spatial zones. His 0D model with moving reference frame describes three characteristic phases of the oxygen balance in a water parcel travelling downstream from the tidal weir given a constant run-off. In our simulation discharge varies considerably (between ~200 m3/s and ~4000 m3/s) which would vary the distance covered in a certain period by Schroeder's water parcel considerably. Sedimentation of organic material and augmented spatial realism both taken into account by our model lead to substantial modifications the picture described by Schroeder (1997).

We will improve reference to the work of Schroeder (1997) and others to shorten writing on our own results. We will emphasize the novelties revealed by our study, such as hotspots of sedimentation, remineralisation outside the main channel as well as identifying periods when the estuary changes from three-zone to two-zone structure, and vice versa. In the previous submission, line 510, we do already mention that the three-fold structure is lost at times of high discharge. To our best knowledge previous modelling studies on Elbe estuary coupled physics and biogeochemistry have not revealed such details of seasonal nitrogen dynamics.

**Line 458: where is fig 12e?**

Authors: We change Fig. 12e as Fig. 12a, which shows the nitrate concentration at the station inside the harbor basin.

**Page 15 Line 478 – Page 16 Line 512 What is new &relevant in this section?**

Authors: This section describes the effect of inter-annual variability of the forcing onto the estuarine ecosystem taking into account the increased in comparison with earlier studies resolution of the complex channel geometry and associated increased spatial resolution of the coupled physical-biogeochemical processes. The ecosystem response to river discharge is very relevant both to scientific discussions as well as management and policy makers. We see both novelty and relevance given.

Page 16 Chapter Conclusions Line 514 –522: Everything that is mentioned here has been shown already several times, for the Elbe and for other estuaries. A 3D model that is able to simulate all these processes is also not new but more or less common (see the comment on the abstract).

Authors: We agree that lines 514-520 do not well represent the focus of our paper. We clearly say in line 521 that we reproduce the general findings of others - this is simply necessary to go beyond what others have done in a credibly way. The patterns of biogeochemical variability

in the area of Elbe inland delta and the response of the estuarine ecosystem to great variability of riverine discharge like in our simulation of the year 2013 have to the best of our knowledge not been tackled by previous studies.

In the revised manuscript we will better expose he benefits of the unstructured modelling and its relevance in the coupled physical-biogeochemical context. The revealed by our work hot spots of heterotrophic decay in the Elbe inland delta can be a relevant aspect for estuaries worldwide, both in natural configurations or due to port construction. Using 1D, 2D or even structured curvilinear 3D models these processes could not be simulated in a generic way (i.e. without local parametrization of the physics and biogeochemistry) revealing the same systemic features. We will add corresponding statements to the revised manuscript.

**Line 523 – 528: Validation missing**

Authors: The validation is given for the main channel (Figs. 8,9 and S-1, S-2 in initial submission, Tables 3 and 4 and Figures D-4, D-5 below for the revised paper). Our line of argumentation is the following: Our model reproduces the known along-channel patterns of primary producer concentrations and inorganic nitrogen species. In particular the ammonium maximum at approximately km 629 is well predicted by our simulations. We can show (Fig. D-1) that the peak ammonium concentrations emerge outside the main channel and that the along-channel peak is connected to processes in the side-channels and basins. At this point we use the model to reach a justified hypothesis about what is happening in the area of the inland delta. We recommend that dedicated cruises should investigate the processes in small channels and harbor basins. We hope that our modelling study will motivate research effort in this direction. We will add such a statement to the revised paper.

**Line 530 – 540: This section is more a description. What is the important conclusion out of it?**

Authors: This sections wraps up the results of the study. The important conclusion is that most of the time, i.e. given average or near-average river runoff, the system reveals a persistent compartmentation. However, the hindcast demonstrates that during the June 2013 flood event the usual and known compartmentation collapses being restored in the aftermath of the flood event.

**Line 542 – 543: This has already been shown for different estuarine systems worldwide.**

Authors: We will provide information on related studies in the revised manuscript. We will also better expose the relevance of using an unstructured 3D model and the importance of resolving narrow channels and complex channel geometry in estuarine ecosystem modelling.

Line 543: From my point of view the importance of the harbor is the most interesting aspect that remains. However, I miss: 1. Better validation, 2. more focus, 3. analyzing hydrodynamic aspects as well (e.g. larger water volume in the harbor compared to section upstream), 4. comparison to other similar systems (like Guadalquivir-Sevilla, Loire, Humber or other estuaries containing large ports).

Authors: We are grateful for this comment. We agree that resolving the Elbe inland delta including the harbor represents a key novelty of our modelling study. We will streamline the manuscript to better present this key aspect of the work. This will be partially achieved by

reducing number of figures on validation and replace them with tables. We will separate result and discussion sections such that a separate discussion section will focus on novel results associated with unstructured modeling and resolution of small channels etc. The tables 1-4 shown below will be added to the revised paper in order to provide a more comprehensive validation of model performance. We will consider deepened analysis of the hydrodynamical aspects concentrating on the focal area of the study (inland delta region/harbor).

We agree that other systems should be included when discussing modelling results in Elbe estuary. We will add a statement like the one given in our answer on page 1 (last paragraph) herein and further elaborate on common traits and differences.

Figures There are many figures. Some figures (like 8, 9 or especially 12) are not easy to understand. Some panels (especially Figure 13) are very small. Check for Figure 2(b) to (e) if the description of x and y axis and the conclusion that in the model salinity intrudes too far into the estuary is consistent. There is no white dashed line visible in Figure 10.

We will reduce complexity of figures 8,9 and 12. Figs. 8 and 9 will be changed for Fig. D-4 below. Complexity of Fig. 12 will reduced by showing less graphs in Fig. 12a and removing Fig. 12d which is out of focus. We will increase panel size were needed (Fig. 13) and add white dashed line to panels in Fig. 10. Information contained in Figs. 4-7 will be mainly moved to Table 3 (please see below) and time series of biogeochemical variables will be shown for the station in the focal area only (Fig. D-5 below).

Technical correction There is a plenty of typing errors, careless mistakes and wrong word order, some obviously due to copy& paste. I'm not a native English speaker, and therefore I'm not in the position to give suggestions for language corrections. However, I have the impression that language, including grammar and use of uniform tenses, needs substantial improvement.

We apologize for the mistakes made. We will consider correction by a native speaker of the manuscript text prior to submitting the revised version of the manuscript in order to improve the language.

**Additional references:**

Arashkevich, E. G., Flint, M. V., Nikishina, A. B., Pasternak, A. F., Timonin, A. G., Vasilieva, J. V., Mosharov, S.A., and Soloviev, K. A. (2010). The role of zooplankton in the transformation of the organic matter in the Ob estuary, on the shelf, and in the deep regions of the Kara Sea. *Oceanology*, *50*(5), 780-792.

Fringer, O. B., Dawson, C. N., He, R., Ralston, D. K., & Zhang, Y. J. (2019). The future of coastal and estuarine modeling: Findings from a workshop. *Ocean Modelling*, 101458.

Graham, J. A., O'Dea, E., Holt, J., Polton, J., Hewitt, H. T., Furner, R., Guihou, K., Brereton, A., Arnold, A., Wakelin, S., Castillo Sanchez, J. M., and Mayorga Adame, G. (2018). AMM15: a new high-resolution NEMO configuration for operational simulation of the European north-west shelf. *Geoscientific Model Development*, *11*(2), 681-696.

Hillebrand, G., Hardenbicker, P., Fischer, H., Otto, W., and Vollmer, S. (2018). Dynamics of total suspended matter and phytoplankton loads in the river Elbe. *Journal of soils and sediments*, *18*(10), 3104-3113.

Kim, T., Sheng, Y. P., & Park, K. (2010). Modeling water quality and hypoxia dynamics in Upper Charlotte Harbor, Florida, USA during 2000. *Estuarine, Coastal and Shelf Science*, *90*(4), 250-263.

Lajaunie-Salla, K., Wild-Allen, K., Sottolichio, A., Thouvenin, B., Litrico, X., & Abril, G. (2017). Impact of urban effluents on summer hypoxia in the highly turbid Gironde Estuary, applying a 3D model coupling hydrodynamics, sediment transport and biogeochemical processes. *Journal of Marine Systems*, *174*, 89-105.

Muylaert, K., Van Wichelen, J., Sabbe, K., & Vyverman, W. (2001). Effects of freshets on phyto-plankton dynamics in a freshwater tidal estuary (Schelde, Belgium). *Archiv fur Hydrobiologie*, *150*(2), 269-288.

Ralston, D. K., Cowles, G. W., Geyer, W. R., & Holleman, R. C. (2017). Turbulent and numerical mixing in a salt wedge estuary: Dependence on grid resolution, bottom roughness, and turbulence closure. *Journal of Geophysical Research: Oceans*, *122*(1), 692-712.

Ruiz, J., Macías, D., Losada, M. A., Díez-Minguito, M., & Prieto, L. (2013). A simple biogeochemical model for estuaries with high sediment loads: Application to the Guadalquivir River (SW Iberia). *Ecological modelling*, *265*, 194-206.

Schöl, A., Blohm, W., Becker, A., Fischer, H. (2008). Untersuchungen zum Rückgang hoher Algenbiomassen im limnischen Abschnitt der Tideelbe. https://www.bafg.de/DE/08\_Ref/U2/01\_mikrobiologie/algen\_tideelbe.pdf?\_\_blob=publication File

Weilbeer, H. (2014). Sediment transport and sediment management in the Elbe estuary. *Die Küste, 81 Modelling*, (81), 409-426.

Wild-Allen, K., & Andrewartha, J. (2016). Connectivity between estuaries influences nutrient transport, cycling and water quality. *Marine chemistry*, *185*, 12-26.

Zhang, J., Zhu, Z., Mo, W. Y., Liu, S. M., Wang, D. R., & Zhang, G. S. (2018). Hypoxia and nutrient dynamics affected by marine aquaculture in a monsoon-regulated tropical coastal lagoon. *Environmental monitoring and assessment*, *190*(11), 656.

|    | km 724 (CUX) |       | km 696 (BRU) |       | km 655 |       | km 636 |       |
|----|--------------|-------|--------------|-------|--------|-------|--------|-------|
|    | Obs.         | Model | Obs.         | Model | Obs.   | Model | Obs.   | Model |
| M2 | 1.37         | 1.40  | 1.26         | 1.29  | 1.32   | 1.35  | 1.49   | 1.47  |
| S2 | 0.34         | 0.36  | 0.29         | 0.31  | 0.29   | 0.31  | 0.33   | 0.34  |
| К2 | 0.21         | 0.22  | 0.19         | 0.20  | 0.20   | 0.21  | 0.22   | 0.22  |
| N2 | 0.10         | 0.09  | 0.08         | 0.08  | 0.09   | 0.08  | 0.10   | 0.08  |
| M4 | 0.12         | 0.07  | 0.14         | 0.11  | 0.10   | 0.10  | 0.13   | 0.12  |
|    | km 629 (HH)  |       | km 623       |       | km 616 |       | km 609 |       |
|    | Obs.         | Model | Obs.         | Model | Obs.   | Model | Obs.   | Model |
| M2 | 1.54         | 1.51  | 1.58         | 1.55  | 1.59   | 1.58  | 1.40   | 1.48  |
| S2 | 0.34         | 0.34  | 0.35         | 0.35  | 0.35   | 0.36  | 0.31   | 0.33  |
| К2 | 0.23         | 0.23  | 0.23         | 0.23  | 0.23   | 0.24  | 0.20   | 0.22  |
| N2 | 0.10         | 0.08  | 0.11         | 0.08  | 0.11   | 0.09  | 0.09   | 0.08  |
| M4 | 0.15         | 0.15  | 0.18         | 0.17  | 0.17   | 0.16  | 0.17   | 0.17  |

Table 1: Tidal amplitudes [m] of five major constitutents are given for observed and simulated elevation at ten gauge stations along the Elbe estuary for the time of model integration (1 June 2012 - 31 December 2013).

| Salinity       | km 725 | km 692 | km 665 | km 651 |        |        |
|----------------|--------|--------|--------|--------|--------|--------|
|                | (CUX)  | (BRU)  |        |        |        |        |
| mean(obs)      | 17.30  | 2.63   | 0.59   | 0.46   |        |        |
| mean(model)    | 21.43  | 9.65   | 2.06   | 0.49   |        |        |
| bias           | 4.13   | 7.02   | 1.47   | 0.03   |        |        |
| R 2 | 0.69   | 0.70   | 0.59   | 0.31   |        |        |
| Wilmott score  | 0.80   | 0.46   | 0.11   | 0.21   |        |        |
|                |        |        |        |        |        |        |
| Temperature    | km 725 | km 692 | km 665 | km 651 | km 629 | km 598 |
|                | (CUX)  | (BRU)  |        |        | (HH)   | (ZOL)  |
| mean(obs)      | 10.23  | 11.92  | 11.76  | 12.70  | 11.49  | 14.45  |
| mean(model)    | 8.18   | 9.06   | 9.22   | 10.42  | 9.49   | 12.38  |
| bias           | -2.05  | -2.86  | -2.54  | -2.28  | -1.99  | -2.07  |
| R 2 | 0.99   | 0.99   | 0.98   | 0.98   | 0.97   | 0.92   |
| Wilmott score  | 0.96   | 0.94   | 0.96   | 0.96   | 0.97   | 0.94   |

Table 2: Observed and simulated average salinity [psu] and water temperature [°C] at several stations along Elbe estuary. The bias. cofficient of determination and Wilmott score for assessment of model performance are also given. The analysis covers the period of model integration between 01 January 2012 and 31 December 2012.

|                | ZOL      | НН       | GRA      | BRU      | CUX      |  |
|----------------|----------|----------|----------|----------|----------|--|
|                | (km 598) | (km 629) | (km 663) | (km 693) | (km 725) |  |
| NO3            |          |          |          |          |          |  |
| Mean(obs.)     | 182.72   | 188.34   | 201.85   | 188.14   | 131.52   |  |
| Mean(model)    | 190.33   | 201.86   | 213.97   | 155.06   | 72.15    |  |
| Bias           | 7.61     | 13.52    | 12.11    | -33.08   | -59.38   |  |
| R 2 | 0.97     | 0.87     | 0.80     | 0.74     | 0.46     |  |
| Willmott Score | 0.99     | 0.94     | 0.92     | 0.86     | 0.64     |  |
| NH4            |          |          |          |          |          |  |
| Mean(obs.)     | 4.73     | 10.20    | 10.75    | 3.13     | 7.60     |  |
| Mean(model)    | 4.31     | 4.32     | 1.77     | 2.29     | 1.56     |  |
| Bias           | -0.42    | -5.88    | -8.98    | -0.84    | -6.05    |  |
| R 2 | 0.02     | 0.29     | 0.01     | 0.03     | 0.06     |  |
| Willmott Score | 0.29     | 0.53     | 0.38     | 0.19     | 0.40     |  |
| PO4            |          |          |          |          |          |  |
| Mean(obs.)     | 1.65     | 1.45     | 1.66     | 2.36     | 1.78     |  |
| Mean(model)    | 1.87     | 2.47     | 4.04     | 3.69     | 1.97     |  |
| Bias           | 0.22     | 1.01     | 2.38     | 1.33     | 0.20     |  |
| R 2 | 0.22     | 0.06     | 0.21     | 0.37     | 0.45     |  |
| Willmott Score | 0.63     | 0.14     | 0.17     | 0.35     | 0.63     |  |
| SiO4           |          |          |          |          |          |  |
| Mean(obs.)     | 70.97    | 68.55    | 69.15    | 41.09    | 45.85    |  |
| Mean(model)    | 73.14    | 60.45    | 65.33    | 43.71    | 23.71    |  |
| Bias           | 2.18     | -8.10    | -3.82    | 2.62     | -22.15   |  |
| R 2 | 0.89     | 0.92     | 0.93     | 0.19     | 0.73     |  |
| Willmott Score | 0.97     | 0.97     | 0.98     | 0.66     | 0.72     |  |
| 02             |          |          |          |          |          |  |
| Mean(obs.)     | 371.62   | 285.20   | 315.63   | 311.59   | 350.21   |  |
| Mean(model)    | 388.49   | 317.39   | 341.06   | 329.53   | 316.90   |  |
| Bias           | 16.87    | 32.19    | 25.43    | 17.94    | -33.31   |  |
| R 2 | 0.74     | 0.79     | 0.89     | 0.76     | 0.83     |  |
| Willmott Score | 0.90     | 0.91     | 0.95     | 0.91     | 0.86     |  |

Table 3: Observed and simulated average nitrate. ammonia. silicate. phosphate and oxygen concentrations [mmol m-3] at five stations along Elbe estuary (see Fig. 1 for positions). Model performance bias. correlation coefficient and Wilmott score between simulated and observed at *five stations* along the Elbe estuary (ZOL. HH. GRA. BRU. CUX - see Fig. 1 for positions) during the model integration period between 01/01/2012 and 31/12/2013.

| Chla-a         | km 598 (ZOL) | km 609 | km 629 (HH) |
|----------------|--------------|--------|-------------|
| Min(obs)       | 2.10         | 1.40   | 2.62        |
| Min(model)     | 2.27         | 1.91   | 0.00        |
| Max(obs)       | 240.00       | 170.31 | 68.90       |
| Max(model)     | 243.62       | 241.23 | 118.05      |
| Mean(obs)      | 104.94       | 56.83  | 19.21       |
| Mean(model)    | 130.00       | 87.47  | 20.54       |
| Bias           | 25.06        | 30.64  | 1.33        |
| R 2 | 0.83         | 0.75   | 0.18        |
| Willmott score | 0.93         | 0.82   | 0.57        |

Table 4: Ranges. bias. correlation coefficient and Wilmott score between observed Chla-a concentration and derived from simulated primary producer concentrations Chl-a concentrations at *three stations* along the Elbe estuary (see Fig. 1 for positions) during the period of model integration between 01/01/2012 and 31/12/2013.

**Fig. D-1:** Vertically averaged and time-averaged for the period of 1 June 2012 until 30 September 2012 model simulated concentrations of (a) meso-zooplankton, (b) particulate nitrogen (PON), (d) ammonia, as well as (c) time-averaged near-bottom oxygen saturation in the area of the Elbe inland delta, where (a), (b) and (d) are given in units of [mmol N m-3]. The location of two measuring stations is also given: The black triangle and the black circle mark the location of chlorophyll-a measurements at km 609 and km 629, respectively, used for validation in table D-4.

---

## Author Comment (AC4) · 15 Oct 2019

**Answers to Referee 2**

*There are many major issues to this submission:*

We thank the referee for his/her comments. In the following we respond to the individual points. Reviewer's comments are written in italics, while authors' answers are in kept in plain font.

*1) There is no/little hydrodynamic calibration. Authors need consider to submit two manuscripts: one for hydrodynamic and one for water quality dynamics.*

Authors: The Elbe set-up has been derived from a larger set-up of the German Bight (Stanev et al. 2019). The model area of the recent set-up is completely contained in the already published set-up, they share the same topography and – most important – the Elbe set-up is hydro-dynamically driven by the German Bight set-up. A separate publication on the physical estuarine dynamics would be necessary and justified in case the processes involved or parametrization have changed, which is not the case. However, the model integration period has been extended from several months (01.01.2012 to 31.08.2012) to two years, 01.01.2012 to 31.12.2013. Therefore, more observational data has become available for the time of model integration. Following the reviewer's advice we have performed additional validation of water levels, salinity and temperature. Please find tables giving basic statistical assessment in our finalized answers to referee 1 (Tables 1 and 2).

*2) Why authors didn't calibrate the water quality for the bottom part, particularly to the oxygen? Ammonia simulation is a little different from the observed one, any justification?*

Authors: We find the model performance regarding water quality, such as oxygen saturation, is very convincing (please see Fig. 6 of the initial submission, Fig. D-5d and table 3 in the finalized answers to referee 1). In particular the good agreement of simulated oxygen saturation with the observed values particularly at Hamburg station during summertime (Fig. 6c of the initial submission) demonstrates that the predicted by the model increased bottom respiration in this area (shown in Fig. 13b of the initial submission) contributes to the realism of the simulation.

Regarding the ammonia simulation it would be good to know exactly to which "little difference" the reviewer refers to. Fig. 5c, d, f show a mismatch between model and observations. The model underestimating ammonia levels is likely to be due to underestimated water temperature. Therefore the most likely explanation is non-optimal boundary forcing. The ammonia variability is however very similar in model and observations (Fig. 5c, d). The most important dynamical feature is an almost stationary peak of ammonia at the downstream end of the Elbe inland delta. Our model reproduces the shape, magnitude and position of this pattern well in comparison with previous modelling effort in the area (Schroeder, 1997; Holzwarth and Wirtz, 2018). To better illustrate and specify the agreement between model and observations we have performed basic statistical assessment for the stations measuring biogeochemical variables in Figs. 4, 5, 6 and 7 of the intial submission (please find statistic assessment of simulation regarding ammonia and other nutrients in Table 3 added to the finalized answers to referee 1).

*3) The model set up and data description is very weak, and need a lot of work to this part. Again, authors need consider to split this manuscript into two manuscripts. Why choose year 2012 and 2013?*

Authors: We have written in model description (pages 4-5, lines 99-108 and lines 122-141), that the model framework combines two established models, SCHISM (Zhang et al. 2016, Stanev et al. 2019) and ECOSMO2 (Daewel and Schrum, 2013). These models have been

described in details in the previous works. However, there is a novel aspect to this particular set-up which is the coupling through the FABM (Bruggeman and Bolding, 2014). The coupler routine also calls the subroutines prescribing the fluxes between the water column and the organic sedimentary layer. To follow the reviewer's advice, we will add a decent description of the coupling through the FABM:

"The biogeochemical model is implemented in the FABM framework (Bruggeman & Bolding 2014), which is used within the hydrodynamical model SCHISM. The ecosystem model tracers are defined in volume concentration in the grid elements and are transported with the baroclinic simulation. Their internal dynamics of matter cycling and biological production is integrated for each model timestep. The light climate is updated each timestep based on the photoynthetically active radiation at the surface of the water column, the model layers' depth, and the linear combination of exponential extinction of constant estuarine background attenuation and extinction due to organic compounds in the water column. A set of bottom variables at the sea floor is used for a pool of particulate organic matter, which gets filled with settling from the lowermost model layer, resuspended for critical shear stresses, and exchanges oxygen and nutrients during remineralization with the water column above. This bottom layer dynamics and exchange is integrated together with the transport and the pelagic ecosystem dynamics each model timestep. At the open boundaries, the state variables are prescribed for the grid elements at the open boundary. The ecosystem coupler is part of the official SCHISM repository. Both, the coupled SCHISM model as well as the ecosystem model library are part of the GCOAST model system of the Helmholtz-Zentrum Geesthacht."

We will also provide a table with the parametrisations used for the biogeochemical model

About choice of period of simulation: To be consistent with the earlier works including the Elbe estuary (Stanev et al., 2019), we start the simulation in the same year (2012). We integrate it for another year (2013) in order to establish a data-set embracing the seasonal to inter-annual variability of the biogeochemical processes. Furthermore we have observations available for these two years which allowed us to perform the necessary biogeochemical validation. Therefore we find the chosen period well-justified.

*3) This study is very local, and there is no linkage to broad area? What is the contribution of this work the research community? The questions is pretty local, and not novel? Authors even didn't fully answer the questions of introduction part.*

Answers: We agree with the reviewer that linkage with other estuarine studies needs to be improved. Deepened comparison with other biogeochemical studies on estuarine ecosystem and the relevance of our study for the global situation of estuaries will be provided in the revised manuscript.

The novelty of our study is that the unstructured mesh has been used to resolve the 3D coupled physical-biogeochemical processes in the narrow, curved channels and small basins of the Hamburg port area. In order to achieve this in a computationally feasible manor, an unstructured model is a decent tool. Previous modelling studies of the Elbe ecosystem have not aimed at spatially resolving the nitrogen cycle using a realistic geometry (Schroeder, 1997; Holzwarth and Wirtz, 2018). Another new and important feature of our model is the sedimentation and resuspension of nutrients which plays a crucial role in this estuary. Our simulation reveals the hot spots of sedimentation, hypoxia and remineralisation. These are located in particular in the side basins and channels of the harbor area which is why our study reveals a novelty compared to previous modelling studies. In the revised manuscript we will

better clarify the novel aspects of our study. We will also provide its relevance for similar estuarine configurations worldwide.

We agree that the answers to research questions need to be more complete. We find this comment linked to the next one "*however authors want to cover everything*". Therefore, we propose to better streamline the manuscript focusing on the impact of biogeochemical processes in area of the Elbe inland delta onto the estuarine nitrogen cycling.

*4) The mixing diagram was used by Jiang and Xia, 2018 and isn't new. This study is mainly for nitrogen dynamics, however authors want to cover everything. It is a little bit difficult to follow, and authors need think how to make a nice flowchart to this manuscript. Overall, it reads like a modeling or technical report. There are many minor issues, however I would like authors to take care of major issues now.*

We use the mixing diagram to a) validate the along-channel distribution of the inorganic nitrogen species over the estuarine salinity gradient (in isohaline coordinates), b) characterize the mixing behavior of these species along the Elbe River. The method is of course not new (see our references), however there is no such extensive demonstration of mixing behavior of nitrogen species for the Elbe system. As the paper focusses on the nitrogen cycling we consider its mixing behavior a central aspect.

The referee is right that we need better streamlining of the manuscript. As written in the answer to the previous comment, we plan to focus the study onto heterotrophic decay confined to the harbor basins and side channels and its impact onto the estuarine nitrogen cycling. This will lead to a more balanced manuscript, where new scientific inside arising from the more complete representation of processes in the port area will be set in relation to better known spatio-temporal organisation of nutrient cycling in the main channel downstream from the port area.

*Jiang, L. & Xia, M. (2018), Modeling investigation of the nutrient and phytoplankton variability in the Chesapeake Bay outflow plume. Progress in Oceanography, 162, 290-302. doi: https://doi.org/10.1016/j.pocean.2018.03.004*

We thank the referee for recommending this useful reference.